# Short- and long-range roles of UNC-6/Netrin in dorsal-ventral axon guidance *in vivo* in *Caenorhabditis elegans*

**Kelsey M. Hooper**, **Vedant D. Jain**, **Celeste J. Gormly**, **Brian J. Sanderson**, **Erik A. Lundquist**[*]

Department of Molecular Biosciences, Program in Molecular, Cellular, and Developmental Biology, KU Center for Genomics, University of Kansas, Lawrence, Kansas, United States of America

* erikl@ku.edu

## Abstract

Recent studies in vertebrates and *Caenorhabditis elegans* have reshaped models of how the axon guidance cue UNC-6/Netrin functions in dorsal-ventral axon guidance, which was traditionally thought to form a ventral-to-dorsal concentration gradient that was actively sensed by growing axons. In the vertebrate spinal cord, floorplate Netrin1 was shown to be largely dispensable for ventral commissural growth. Rather, short range interactions with Netrin1 on the ventricular zone radial glial stem cells was shown to guide ventral commissural axon growth. In *C. elegans*, analysis of dorsally-migrating growth cones during outgrowth has shown that growth cone polarity of filopodial extension is separable from the extent of growth cone protrusion. Growth cones are first polarized by UNC-6/Netrin, and subsequent regulation of protrusion by UNC-6/Netrin is based on this earlier-established polarity (the Polarity/Protrusion model). In both cases, short-range or even haptotactic mechanisms are invoked: in vertebrate spinal cord, interactions of growth cones with radial glia expressing Netrin-1; and in *C. elegans,* a potential close-range interaction that polarizes the growth cone. To explore potential short-range and long-range functions of UNC-6/Netrin, a potentially membrane-anchored transmembrane UNC-6 (UNC-6(TM)) was generated by genome editing. *unc-6(tm)* was hypomorphic for dorsal VD/DD axon pathfinding, indicating that it retained some *unc-6* function. Polarity of VD growth cone filopodial protrusion was initially established in *unc-6(tm)*, but was lost as the growth cones migrated away from the *unc-6(tm)* source in the ventral nerve cord. In contrast, ventral guidance of the AVM and PVM axons was equally severe in *unc-6(tm)* and *unc-6(null)*. Together, these results suggest that *unc-6(tm)* retains short-range functions but lacks long-range functions due to reduced secreted UNC-6. Ectopic *unc-6(+)* expression from non-ventral sources did not dramatically perturb dorsal VD growth cone polarity or axon outgrowth, suggesting that ectopic UNC-6 cannot redirect polarity once it is established in the VD/DD neurons. This is not what would be expected of a growth cone dynamically reading a gradient of UNC-6, but is consistent with the Polarity/protrusion model of growth cone guidance away from UNC-6/Netrin.

**Data availability statement:** Strains and plasmids are available upon request. FASTQ files for this project are available in the Sequence Read Archive, Project Numbers PRJNA916208 and PRJNA1093133. All other relevant data are in the manuscript and its supporting information files.

**Funding:** This work was supported by the National Institutes of Health (R03 NS114554, R56 NS095682, and R01 NS115467 to EAL). The funders had no role in study design, data collection and analysis, decision to publish, or preparation of the manuscript.

**Competing interests:** The authors have declared that no competing interests exist.

## Author summary

In the developing nervous system, neurons extend thin cellular processes called axons that make precise contacts with other neurons in elaborate networks. The structure of these axon networks underlies the function of the nervous system. Thus precise axon guidance is critical during development to produce a functional nervous system. Axons are guided by the growth cone, a dynamic structure at the distal tup of the axon that senses and responds to signals in the environment that instruct axon guidance. One such signal if UNC-6/Netrin, a conserved, laminin-like secreted molecule that guides many aspects of axon guidance. In the nematode *Caenorhabditis elegans*, UNC-6 controls dorsal-ventral axon guidance of motor neurons and sensory neurons. The canonical textbook model of UNC-6/Netrin function involved the formation of a gradient of secreted UNC-6/Netrin in the ventral-to-dorsal axis which growth cones dynamically sensed and responded by growing up or down the gradient. The gradient model does not explain recent studies in vertebrates and *C. elegans* that have shown close-range or contact-mediated events that more resemble cell polarity events than dynamic reading of a gradient. In *C. elegans*, the growth cone of the VD axon is polarized dorsally by UNC-6/Netrin. UNC-6/Netrin then maintains this polarity and regulates protrusion based upon this polarity, stimulating protrusion dorsally and inhibiting protrusion ventrally, resulting in net dorsal outgrowth. This Polarity/Protrusion model resembles events that take place in ventral growth of sensory axons, the Statistically-Oriented Asymmetric Localization (SOAL) model. The Polarity/Protrusion model predicts that a short-range UNC-6 signal polarizes the VD growth cone, and that secreted, diffusible UNC-6/Netrin maintains initial polarity and regulates protrusion. This model is critically tested here by the construction of a membrane-anchored UNC-6 genome edit predicted to not produce any diffusible UNC-6, thus blocking any long-range function. Data here show that VD growth cones are initially polarized corerectly in *unc-6(tm)*, but that this polarity is lost as the growth cones move dorsally away from the UNC-6(TM) source. This is consistent with a short-range interaction that polarizes the growth cone, and a long-range function needed to maintain polarity. Furthermore, ectopic expression of diffusible UNC-6 from ectopic sources did not perturb VD growth cone polarity, suggesting that the initial polarity event cannot be altered by later diffusible UNC-6/Netrin, as would be expected with a gradient model. These results are consistent with the Polarity/Protrusion model of UNC-6 function.

## Introduction

UNC-6/Netrin is a conserved regulator of dorsal-ventral axon guidance [1–4](reviewed in [5]). It was previously thought that the UNC-40/DCC receptor mediated ventral growth and attraction to the Netrin source [6–9], whereas the UNC-5 receptor mediated dorsal growth and repulsion from the Netrin source [10,11]. Recent studies of UNC-6/Netrin on the growth cones of axons *in vivo* in *C. elegans* show that UNC-40/DCC and UNC-5 are each involved in both ventral and dorsal guidance. In the HSN neuron, which extends an axon ventrally, the protrusive activity of UNC-40 is polarized ventrally by UNC-6, and refined and maintained by UNC-5 (the Statistically-Oriented Asymmetric Localization (SOAL) model) [12–14]. In VD growth cones, which grow dorsally, UNC-6/netrin first polarizes the growth cone via UNC-5, such that UNC-40 protrusive activity is localized dorsally, and then maintains this polarity

and regulates growth cone protrusion [4,15,16]. UNC-6 drives protrusion dorsally through UNC-40, and inhibits protrusion ventrally and laterally through UNC-5, resulting in net dorsal outgrowth (the Polarity/Protrusion model) [4,15–17]. Common to the SOAL and Polarity/Protrusion models is the polarization of pro-protrusive UNC-40/DCC activity by UNC-6 and UNC-5, which results in directed protrusion and growth cone advance. In ventral growth (SOAL), UNC-6 polarizes UNC-40 toward the ventral UNC-6/netrin source, and in dorsal growth, UNC-6 polarizes UNC-40 activity away from the ventral source.

At the time of VD growth cone outgrowth, UNC-6 is expressed in the VA and VB neurons in the ventral nerve cord, whose processes extend the length of the ventral nerve cord [18]. The dorsally-directed VD/DD motor axons also extend in the ventral nerve cord in proximity to the VA/VB processes. In contrast, the HSN and AVM/PVM neuron cell bodies reside laterally and extend processes ventrally toward the ventral nerve cord. Thus, UNC-6 might have both short- and long-range functions, the later relying on diffusion of UNC-6 away from its ventral expression source. In the vertebrate spinal cord, a long-range function of Netrin1 expression in the floorplate was invoked to explain ventral guidance of dorsal commissural axons [19–21]. However, in *Drosophila*, Netrin-1 was shown to have close-range roles independent of DCC [22]. Further, cell-specific knock-out experiments showed that floorplate Netrin1 was largely dispensable for commissural axon guidance [23–26]. Instead, Netrin1 expression on the radial glial ependymal cells was required, in a short-range, possibly contact-mediated process. Thus, vertebrate Netrin1 might also have short-range functions.

In the Polarity/Protrusion hypothesis of VD dorsal growth cone guidance, a short-range interaction with UNC-6 in the ventral nerve cord polarizes the growth cone, and a longer-range role dependent upon diffusible UNC-6 maintains this polarity and regulates extent of protrusion biased dorsally [4,15–17]. In this model, it is predicted that only the initial, short-range interaction is required to come from a directional source (ventral), whereas the long-range function is predicted to not require a directional source to maintain polarity and regulate protrusion.

To analyze potential short- and long-range roles of UNC-6/Netrin, the endogenous *unc-6* gene was edited such that a transmembrane domain was introduced at the C-terminus of the UNC-6 protein (the *unc-6(lq154)* genome edit, or *unc-6(tm)*). Transmembrane UNC-6 (UNC-6(TM)) is predicted to be anchored in the plasma membrane of the cell in which it is expressed. *unc-6(tm)* was hypomorphic, and animals were less severely uncoordinated and had fewer VD/DD axon guidance defects compared to the *unc-6(ev400)* null. VD growth cones were initially properly polarized in *unc-6(tm)*, but polarity was lost as the growth cones advanced dorsally away from the ventral nerve cord. In contrast *unc-6(null)* growth cones were unpolarized as soon as they emerged from the ventral nerve cord. This suggests that UNC-6(TM) is sufficient to initially polarize the VD growth cone (at short range) but that diffusible UNC-6 is required to maintain polarity (long-range). The ventral guidance of the AVM and PVM axons in *unc-6(tm)* resembled the *unc-6(null)*, consistent with *unc-6(tm)* lacking long-range UNC-6 activity.

The Polarity/Protrusion model predicts that maintenance of VD growth cone polarity would not require a directional source of UNC-6 once established by a short-range interaction. To test this idea, *unc-6(+)* was expressed ectopically in the pharynx in the anterior and in dorsal body wall muscle. In a wild-type background, ectopic *unc-6(+)* expression had no effect on VD/DD axon guidance or VD growth cone polarity, consistent with the Polarity/Protrusion model but not consistent with a growth cone dynamically reading a gradient of UNC-6. Ectopic UNC-6 did weakly perturb AVM/PVM ventral guidance, suggesting that axons can be affected differently by ectopic UNC-6 in distinct contexts, in this case those near the source of UNC-6 (VD/DD) or at a distance from it (AVM/PVM).

## Results

### *unc-6(lq154)* can encode a transmembrane UNC-6 molecule

Cas9 genome editing was used to create *unc-6(lq154)* (also called *unc-6(tm)*), which has the potential to encode a transmembrane UNC-6 molecule (UNC-6(TM)) (Fig 1A). Intron 12 and exon 13 were replaced by a synthetic mini-intron and a recoded exon 13 that retained the identical coding potential of wild-type exon 13 but lacked the stop codon. *Green fluorescent protein (gfp)* coding region without a stop codon was fused in frame to exon 13. This was followed by a synthetic intron containing the *Hygromycin-resistance* (*HygR*) selection cassette in the opposite antiparallel direction, and a new exon containing the coding region for the transmembrane domain of the *cdh-3* gene fused in frame to *gfp*. The genomic sequence of *unc-6(lq154)* was confirmed by sequencing overlapping PCR products across the region. The sequence of the repair plasmid used to create *unc-6(lq154)*, which is the sequence of the *unc-6(lq154)* mutation, can be found in S1 Data.

The *HygR* gene was flanked by *LoxP* sites, but experiments to remove the *LoxP* region using *Cre* expressed in the germline were unsuccessful. To assess transcripts from the genome-edited region, RNA-seq from *unc-6(lq154)* was conducted. Reads were aligned to a reference genome file that was edited to include the *unc-6(lq154)* sequence (S2 Data). Reads from transcripts spanning the removal of the synthetic mini-intron and the *HygR* selection cassette intron were identified (Fig 1B). However, a common splicing pattern included an unpredicted novel exon (the linker exon) within the *HygR* cassette (Fig 1A and 1B). This linker exon is 150 bp in length and retains the reading frame from *gfp* to the transmembrane domain-containing final exon. Furthermore, it includes no in-frame stop codons. The net result of inclusion of the exon is an additional 50 amino acids between GFP and the transmembrane region with no detectable similarity to other proteins (Fig 1C). In many cases, the introns flanking the linker exon were not removed, resulting in in-frame premature stop codons in each case, and likely nonsense-mediated decay. However, it is possible that *unc-6(lq154)* produces some secreted, diffusible UNC-6. If it does, not enough is produced to mediate all roles of *unc-6*, as *unc-6(lq154)* mutants are uncoordinated and display axon guidance defects (see below). UNC-6 In sum, *unc-6(lq154)* is predicted to encode transmembrane UNC-6 molecules with and without the 50 extra linker amino acids. The potential UNC-6(TM) protein is predicted to be a trans-membrane protein using the Trans Membrane Hidden Markov Model (TMHMM) algorithm (Fig 1D).

It is expected that UNC-6(TM)::GFP would remain associated with the cell expressing it and not diffuse. Despite the *gfp* tag, no UNC-6(TM)::GFP fluorescence could be detected in *unc-6(lq154)* animals. Furthermore, no transgenic UNC-6::mVenus expression was detected using the previously-described *Punc-6::unc-6::mVenus* transgene [27], despite this transgene fully rescuing *unc-6* mutants (see below). This could be due to a low level of expression from the endogenous *unc-6* locus, dynamic expression, or both. It is also possible that the GFP tag does not function efficiently in the extracellular space.

Previous studies detected endogenous UNC-6 protein using immunofluorescence [18]. In these studies, UNC-6 immunofluorescence appeared to localize to cell bodies from which it was expressed. Likewise, transgenic expression of UNC-6::mVenus detected using immu-nofluorescence appeared to localize to the cell bodies from which it was expressed [27]. In both cases, it was difficult to conclude with confidence that extracellular UNC-6 that had diffused from the site of expression was detected. UNC-6 clearly functions at a distance (*e.g.,* in AVM/PVM and HSN axon guidance [12–14,28]). The inability to confidently detect diffusing UNC-6 could be due to the low levels and dispersed distribution of UNC-6 in the extracellular space.

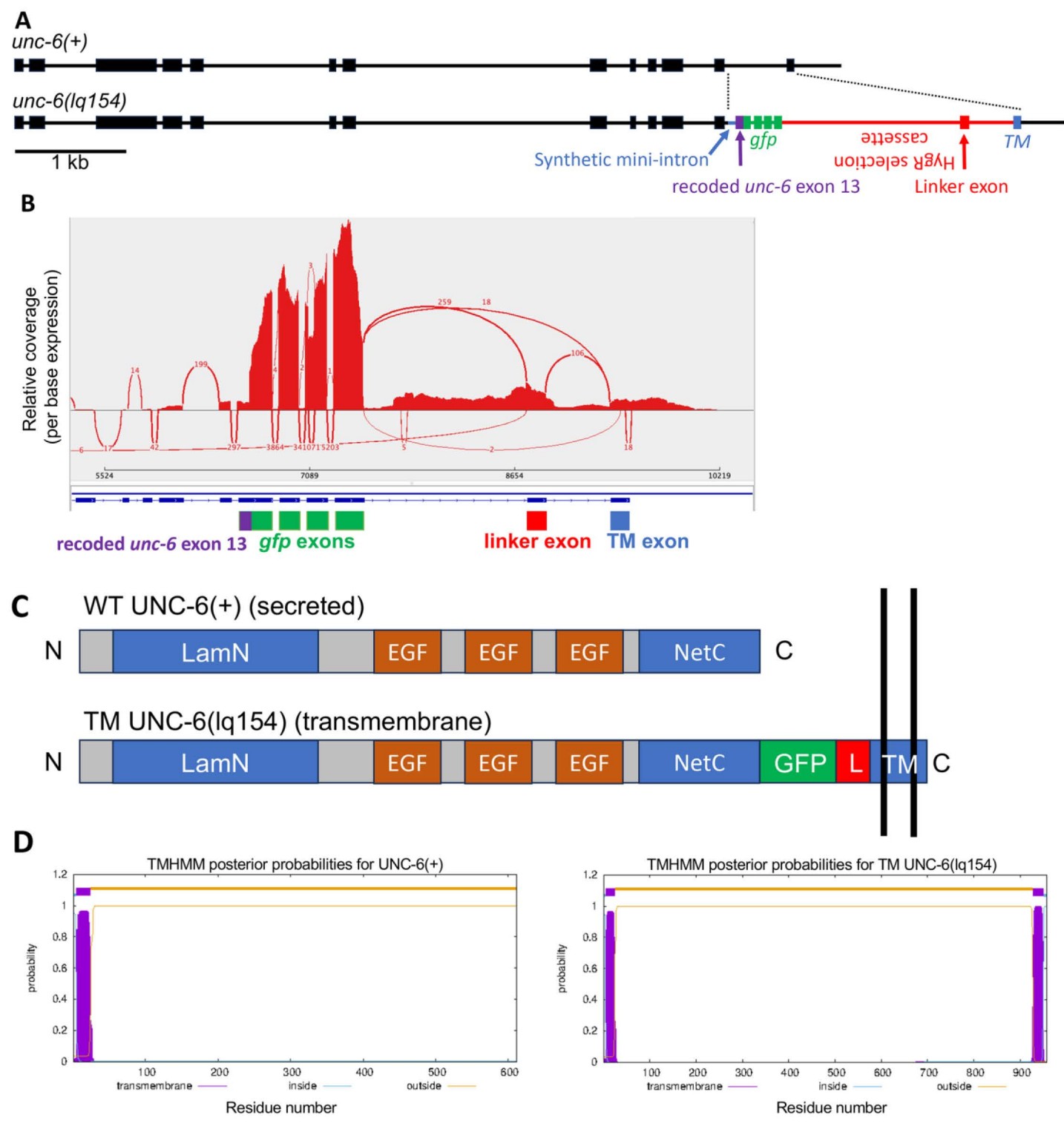

**Fig 1. *unc-6(lq154)* genome edit structure.** A) The *unc-6(+)* and *unc-6(lq154)* gene structures. Boxes represent exons and lines represent introns. In *unc-6(lq154)*, intron 12 and exon 13 (dashed line) were replaced by: a synthetic mini intron (blue), a recoded exon 13 lacking a stop codon but with the identical coding potential as *unc-6(+)* (purple), *green fluorescent protein* in frame with *unc-6* (green), a *Hygromycin-resistance* cassette, with the *HygR* gene on the opposite strand (red), an exon encoding the transmembrane domain (TM) from the *cdh-3* gene with a stop codon (blue), followed by the endogenous *unc-6* 3' UTR region. B) A Sashimi plot from the Integrated Genome Viewer of RNA-seq reads from *unc-6(lq154)* aligned to a reference genome edited to include the *unc-6(lq154)* genome edit. The gfp to linker exon splice occurred 18 times, but the predominant splice variant (259) included a novel, likely artifactual, exon in the *HygR* cassette (the linker exon). This exon is 150-bp

long and when translated, is in frame with *gfp* and the TM exon and contains no in-frame stop codons. The exon adds an additional 50 amino acid residues to the molecule with no significant similarity to other proteins. In many cases, the introns flanking the linker exon were not removed, resulting in in-frame premature stop codons in each case. Read coverage of *gfp* exons was greater than that of surrounding *unc-6(lq154)* sequence because the strain used for RNA-seq included the *Punc-25*::gfp transgene *(juIs76)*. C) A diagram of the UNC-6(+) and UNC-6(lq154) TM molecules. LamN is the laminin N domain; EGF are epidermal growth factor repeat domains; NetC is the Netrin C-terminal domain; GFP is green fluorescent protein; L is the linker exon; and TM is the transmembrane domain. UNC-6 is secreted, and UNC-6(lq154) is predicted to be a transmembrane protein. D) Trans Membrane Hidden Markov Model (TMHMM) analysis of UNC-6(+) predicts a secreted protein with an N-terminal signal peptide, and that UNC-6(lq154) encodes a transmembrane protein with an N-terminal signal peptide and a C-terminal transmembrane region.

### *unc-6(lq154tm)* is hypomorphic for dorsal VD/DD axon guidance

The *unc-6(ev400)* null mutants are severely uncoordinated and display severe defects in the ventral to dorsal guidance of the VD and DD GABA-ergic motor axons. The VD/DD cell bodies reside in the ventral nerve cord. Axons extend anteriorly in the ventral nerve cord, and then turn dorsally for commissural migration (Fig 2A and 2D). In wild-type, an average of 16 VD/DD ventral-to-dorsal commissures are apparent. In *unc-6(ev400)* null mutants, an average of 10 commissures showed any obvious extension at all from the ventral nerve cord (Fig 2F and 2G), an average of 3.2 crossed the lateral midline (Fig 2H), and an average of fewer than one commissure per animal reached the dorsal nerve cord (Fig 2I).

 *unc-6(lq154)* animals were less severely uncoordinated than *unc-6(ev400)* (Fig 2B and 2C). *unc-6(ev400)* animals are uncoordinated, with kinked body posture, and tend to not move and stay clustered in one place. *unc-6(lq154)* are less uncoordinated, show a more defined S-shaped body posture and movement, and tend the move and spread more evenly across the lawn of *E. coli*. *unc-6(lq154)* displayed VD/DD axon guidance defects (Fig 2E), but significantly fewer compared to *unc-6(ev400)* null mutants: significantly more commissures emanated from the ventral nerve cord (15 compared to 10) (Fig 2G); significantly more crossed the lateral midline (7.8 compared to 3.2) (Fig 2H), and significantly more reached the dorsal nerve cord (2 compared to <1) (Fig 2I). These data indicate that *unc-6(lq154)* is not a complete loss of *unc-6* function and is hypomorphic for dorsal VD/DD axon guidance.

### *unc-6(lq154tm)* resembles the *unc-6* null phenotype for AVM/PVM ventral axon guidance

The AVM and PVM mechanosensory neurons reside laterally and are positioned dorsally away from the ventral nerve cord (Fig 3A, 3B and 3E). AVM and PVM axon extend directly from the ventral region of the cell body and extend ventrally to the ventral nerve cord. *unc-6(ev400)* null mutants display 72% and 73% failure of AVM and PVM ventral guidance, respectively (Fig 3C–3I). The misguided axons extended anteriorly and failed to reach the ventral nerve cord. *unc-6(lq154)* AVM and PVM ventral guidance did not differ significantly from *unc-6(ev400)* (63% and 62%) (Fig 3H and 3I). These axon guidance data suggest that *unc-6(lq154)* might more strongly affect the long-range guidance of axons toward the ventral nerve cord.

### VD growth cones were initially polarized dorsally in *unc-6(lq154)*

The hypomorphic VD/DD dorsal axon guidance phenotype of *unc-6(lq154)* was investigated in more detail by analyzing VD growth cone polarity during their dorsal commissural outgrowth. VD growth cones in wild-type display a dorsal polarity of filopodial protrusions, such that ~70% of filopodial protrusions occur from the dorsal half of the growth cone [4,29] (Fig 4A–4C). This dorsal polarity of filopodial protrusion is abolished in *unc-6(ev400)* mutants, indicating that *unc-6* is required to polarize the VD growth cone during outgrowth [4] (Fig 4B and 4F).

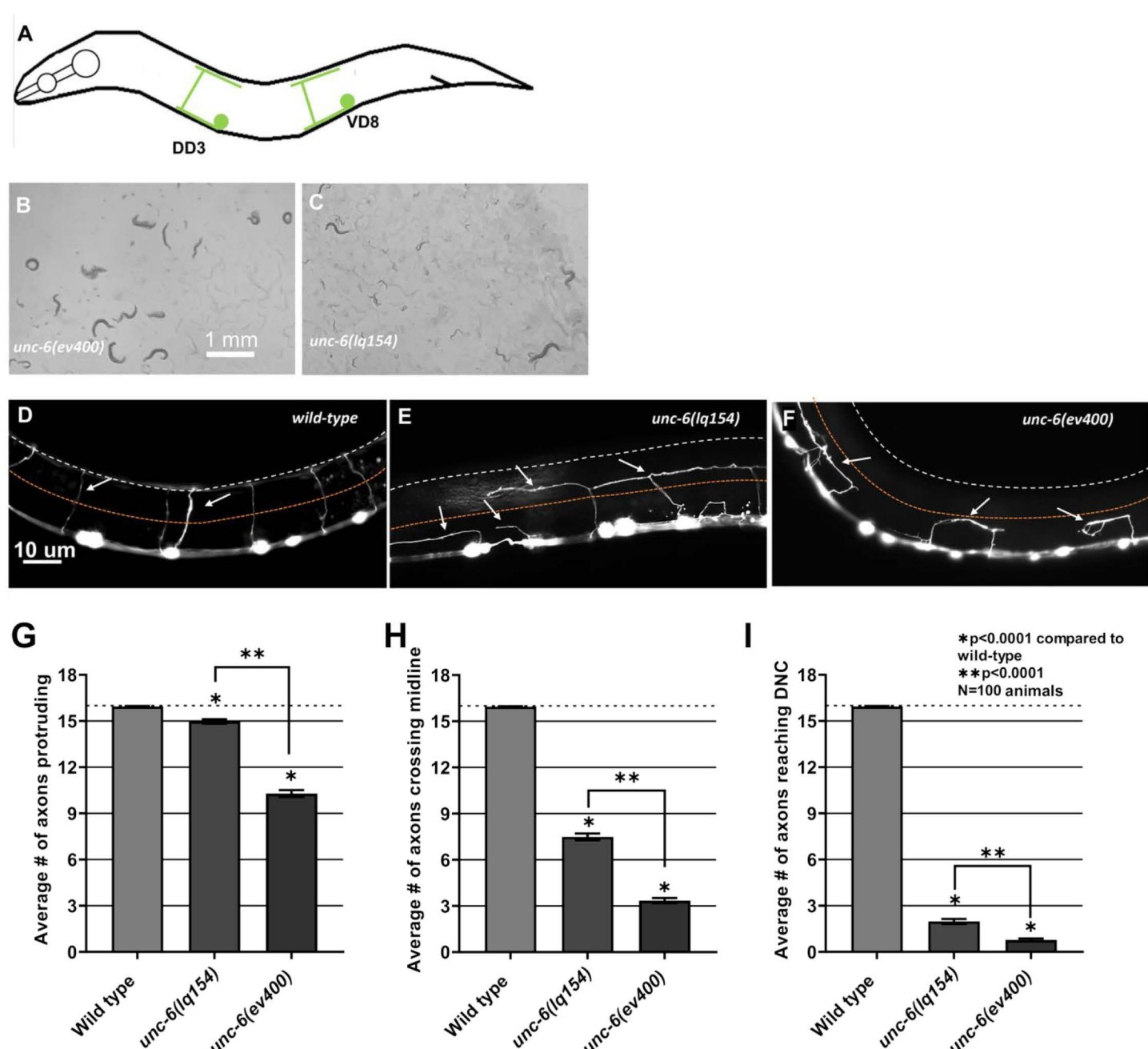

**Fig 2. VD/DD axon guidance defects in *unc-6* mutants.** A) A diagram showing the structure of the VD and DD axons. Anterior is to the left and dorsal is up. Only one of 13 VDs and one of 6 DDs is shown. Cell bodies reside in the ventral nerve cord. Axons extend anteriorly in the VNC, and then turn dorsally to migrate commissurally to the dorsal nerve cord, where they turn posteriorly and extend in the DNC, tiling the VNC and DNC. B, C) Light micrographs of *unc-6(ev400)* and *unc-6(lq154)* animals in lawns of *E. coli* on NGM plates. The scale bar in B represents 1 mm. D–F) Fluorescent micrographs of VD/DD commissural axons in wild-type and *unc-6* mutants expressing the *juIs76[Punc-25::gfp]* transgene. The lateral midline is indicated by a dashed orange line and the dorsal nerve cord is indicated by a dashed white line. White arrows point to axons. The scale bar in D represents 10 μm for all images. G–I) Graphs quantifying VD/DD axon pathfinding defects in *unc-6* mutants. The X-axis represents genotype and the Y axis the number of commissures observed in animals. In wild-type, an average of 16 commissures extend from the ventral nerve cord to the dorsal nerve cord. Error bars represent standard error of the mean. Statistical significance was determined using a two-tailed *t*-test with unequal variance. G) The average number of commissures emanating from the ventral nerve cord. H) The average number of commissures extending past the lateral midline. I) The average number of commissures reaching the DNC.

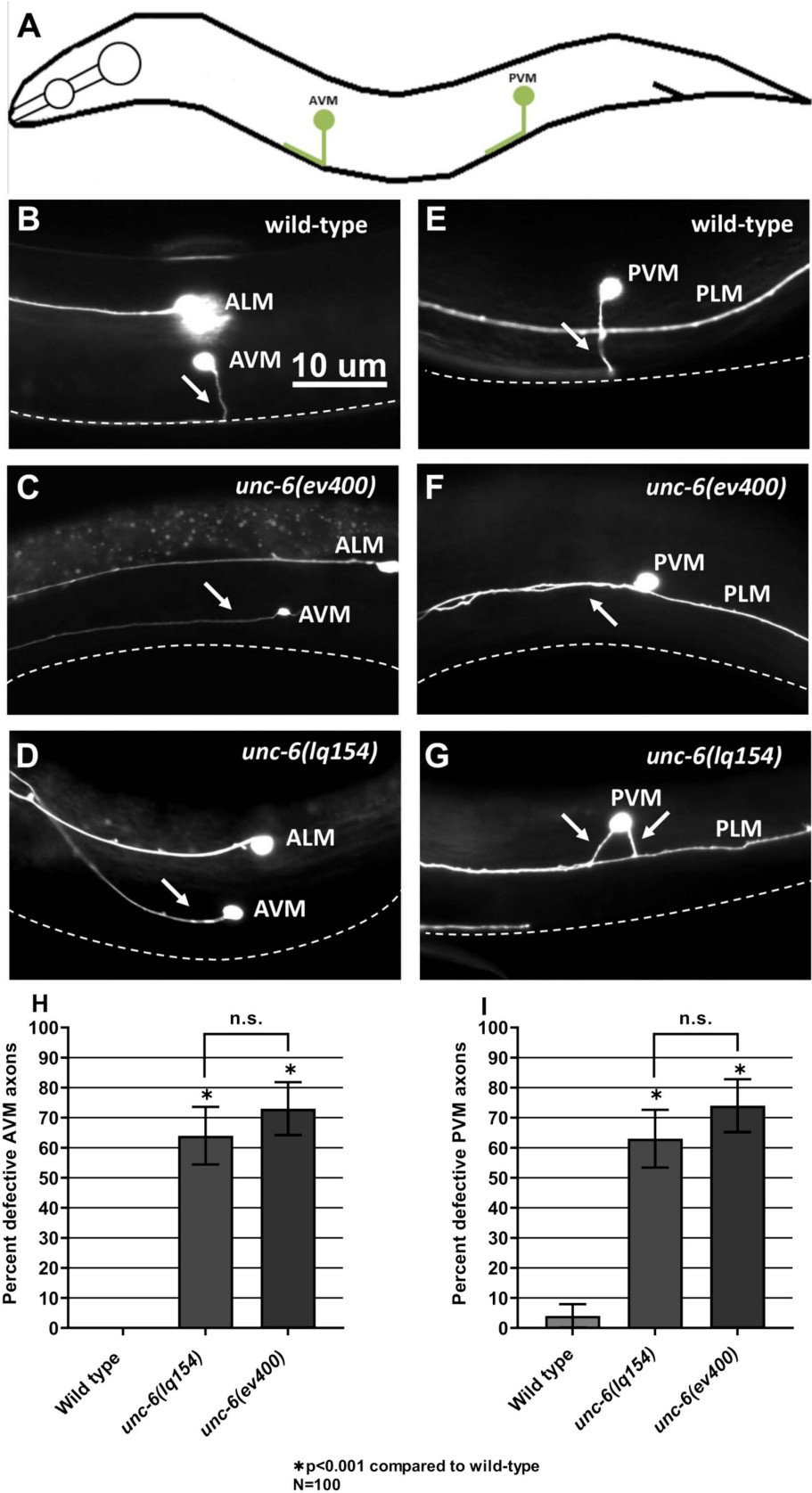

**Fig 3. AVM/PVM axon guidance defects in *unc-6* mutants.** A) A diagram illustrating AVM and PVM structure. Anterior is to the left and dorsal is up. The AVM and PVM cell bodies are located laterally and extend axons that

migrate ventrally to join and run anteriorly within the ventral nerve cord. B, C) Fluorescent micrographs of AVM and PVM neurons expressing the *zdIs4[Pmec-4::gfp]* transgene in wild-type and *unc-6* mutants. The ALM and PLM cell bodies and processes are also indicated. The ventral nerve cord is indicated by a white dashed line. In *unc-6* mutants, AVM and PVM axons fail to extend ventrally and instead extend laterally, failing to reach the VNC. The scale bar in B represents 10 μm for all images. H, I). Graphs showing the percentage of defective AVM and PVM axons (failure to reach the ventral nerve cord, branching, or turning at >45° angle during migration) in *unc-6* mutants. Error bars represent 2× the standard error of proportion. Significance was determined using Fisher's exact test.

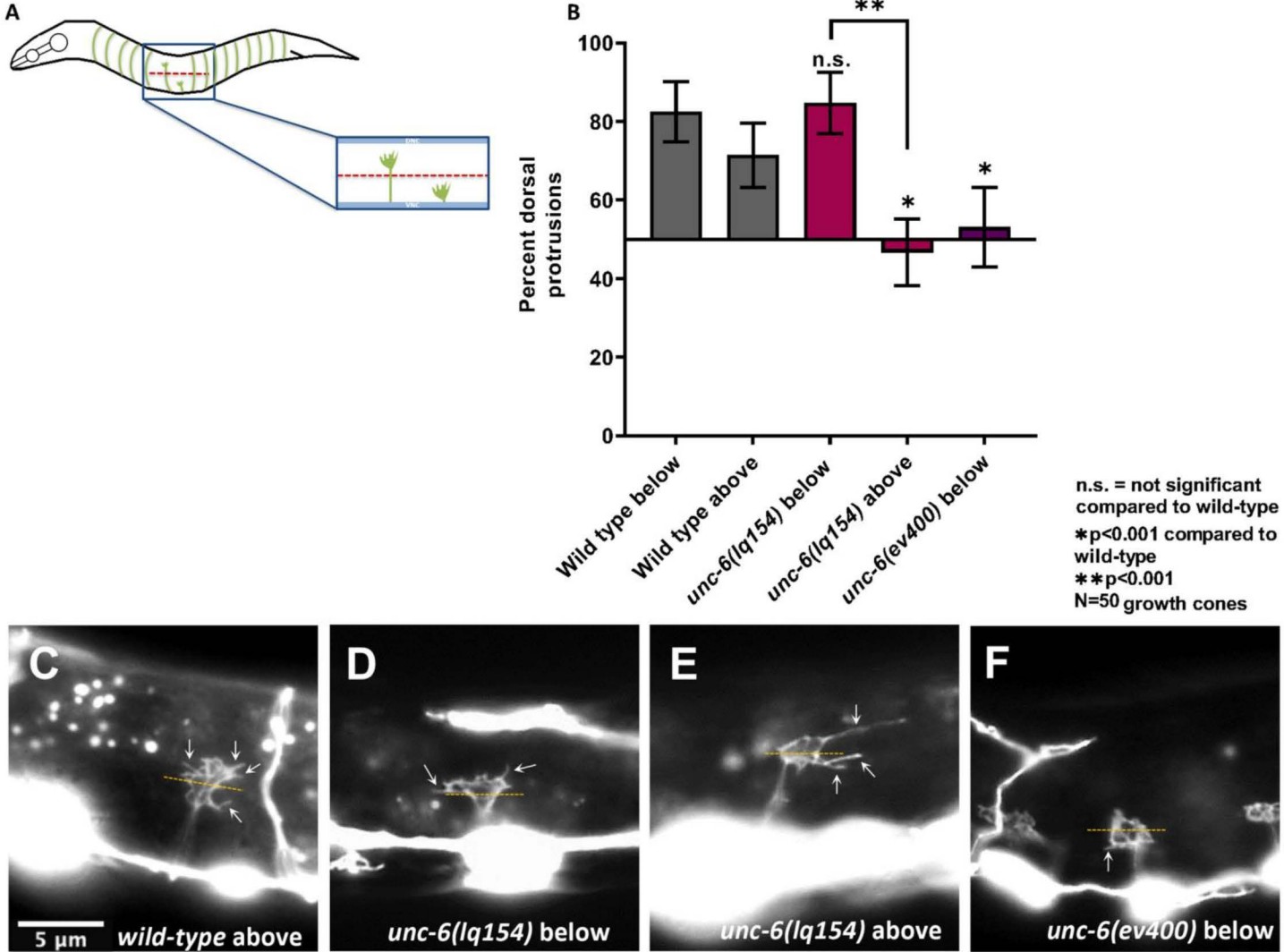

**Fig 4. VD growth cone polarity in *unc-6* mutants.** A) Diagram of early L2 *C. elegans* hermaphrodite illustrating VD/DD axons and VD growth cones. Anterior is to the left and dorsal is up. During early L2 VD/DD axons extend anteriorly in the ventral nerve cord before turning dorsally to extend to the dorsal nerve cord. The lateral midline is represented by the dashed red line and the thick blue lines represent the ventral and dorsal nerve cords. VD growth cones are shown both below and above the midline. B) Percentage of dorsally biased filopodial protrusions of VD growth cones below and above the lateral midline. Significance between wild-type and mutants was determined by Fisher's exact test. Error bars represent 2× the standard error of proportion. C–F) Representative images of VD growth cones expressing the *juIs76[Punc-25::gfp]* transgene. Polarity is determined by dividing the growth cone equally into dorsal and ventral subsections respective to the ventral nerve cord and counting the number of protrusions. The dashed yellow line represents the approximate midline used to divide the growth cone into equal sections. White arrows point to filopodia. The scale bar in C represents 5 μm for all images. C) Wild-type growth cones display dorsally biased filopodial protrusions. D, E) *unc-6(lq154)* does not affect polarity below the midline but disrupts polarity above the midline.

Significantly more VD/DD axons extended past the lateral midline in *unc-6(lq154)* compared to the *unc-6(ev400)* null, suggesting that guidance defects in *unc-6(lq154)* were occurring after lateral midline crossing. Therefore, growth cones at different points in their migration were analyzed, specifically those that were below the lateral midline and those that were above (Fig 4A). Wild-type VD growth cones were dorsally polarized both below and above the lateral midline (82% and 70%, respectively) (Fig 4B and 4C). In *unc-6(ev400)* null mutants, most VD growth cones did not extend past the lateral midline. Those below the lateral midline showed a loss of polarity (52%) (Fig 4B and 4D). In contrast, *unc-6(lq154)* VD growth cones were still significantly polarized below the lateral midline (84%) (Fig 4B and 4D). However, polarity was abolished in *unc-6(lq154)* VD growth cones above the midline (48%) (Fig 4B and 4E). These results suggest that in *unc-6(lq154)*, VD growth cones were initially polarized, but this polarity was not maintained as dorsal outgrowth proceeded. This could explain the hypomorphic nature of *unc-6(lq154)* in dorsal VD axon guidance.

F-actin accumulation is also dorsally-polarized in wild-type VD growth cones, which is abolished in *unc-6(ev400)* null mutants [4] (Fig 5A–5D). F-actin was visualized using the VAB-10 F-actin binding domain fused to GFP as previously described [4]. F-actin was dorsally

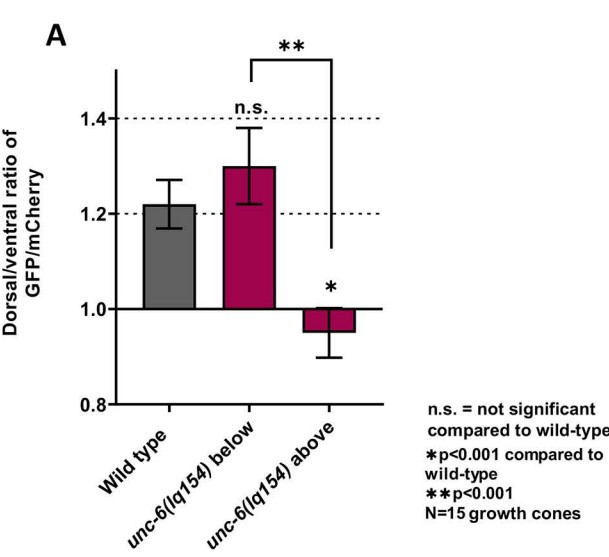

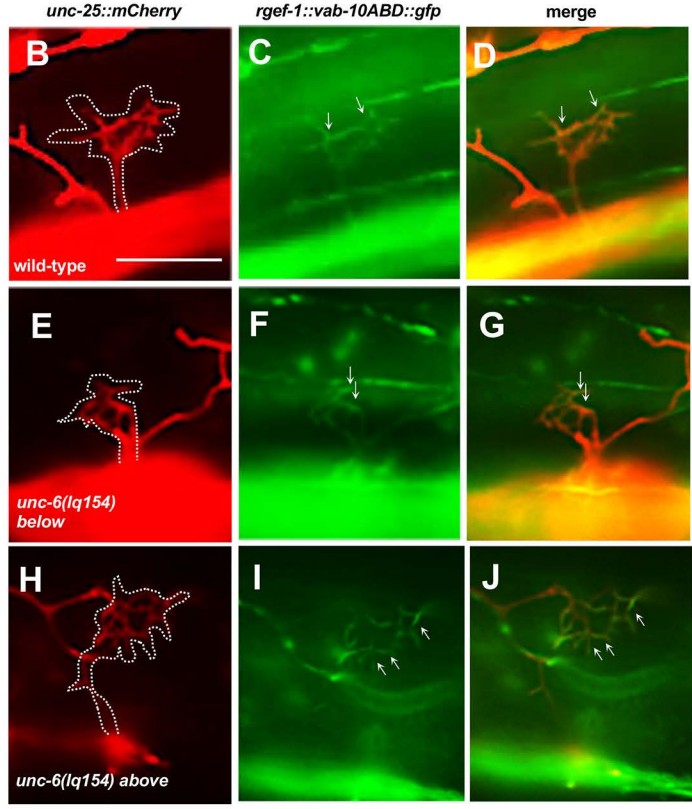

**Fig 5. F-actin polarity is lost above the midline in *unc-6(lq154)*.** A) Graph showing the dorsal-to-ventral ratio of GFP/mCherry in VD growth cones expressing an mCherry volume (*lhIs6[Punc-25::mCherry]* marker and the VAB-10 F-actin binding domain fused to GFP (*lqIs170 [Prgef-1::vab-10ABD:gfp]*). Five line scans from dorsal to ventral were drawn across each growth cone, and the intensities of mCherry and GFP were determined in dorsal and ventral sections. Error bars represent standard error of the mean. Statistical significance was determined using a two-tailed *t*-test with unequal variance. B–J) Images of VD growth cones with mCherry volume marker, VAB-10ABD::GFP, and a merge. The dotted line in the mCherry micrographs represents the approximate area of the growth cone. The scale bar in B represents 5 μm for all images. B–D) A wild-type VD growth cone. Arrows point to dorsally-localized GFP. E–G) An *unc-6(lq154)* growth cone below the lateral midline. Arrows point to dorsally-localized GFP. H–J) An *unc-6(lq154)* growth cone above the lateral midline. Arrows point to ventrally located GFP indicating a loss of dorsal polarity of F-actin.

polarized in *unc-6(lq154)* VD growth cones below the lateral midline (Fig 5A and 5E–5G), but was unpolarized in those above the lateral midline (Fig 5A and 5H–5J). These results further suggest that VD growth cones are initially polarized in *unc-6(lq154)*, and that polarity is lost as the growth cones migrate dorsally.

These results are consistent with *unc-6(lq154)* producing a transmembrane molecule that cannot diffuse, thus not mediating long-range events. However, it is also possible that *unc-6(lq154)* produces some secreted UNC-6 from the transcripts that retain intron sequences upstream of the transmembrane domain in the *unc-6(lq154)* genome edit (Fig 1). If *unc-6(lq154)* does produce some secreted UNC-6, it might not be at levels required to mediate long-range functions of UNC-6 in maintenance of VD polarity or AVM/PVM axon guidance. In fact, this could be why growth cones remain polarized for a distance after leaving the ventral nerve cord. In either case, these results indicate that UNC-6 has both short-range function in establishing VD growth cone polarity and long-range function in maintaining that polarity, and a long-range function in AVM/PVM ventral guidance.

## Ectopic *unc-6* expression did not strongly affect VD/DD dorsal axon guidance

Results with *unc-6(lq154)* suggest that short-range interactions with UNC-6 polarize the growth cone, and that longer range action of UNC-6 is required to maintain growth cone polarity. If a short-range interaction was sufficient to polarize the growth cone, it would be expected that ectopic expression of *unc-6* would not have an effect on VD growth cone polarity or redirect VD/DD axon guidance.

A *Punc-6::unc-6::mVenus* transgene [27], which expresses UNC-6::mVenus from the endogenous *unc-6* promoter, fully rescued the uncoordinated locomotion, VD/DD, and AVM/PVM axon guidance defects of *unc-6(ev400)* and *unc-6(lq154)* mutants (Fig 6). However, no UNC-6::mVenus expression could be detected in these animals likely due to low levels of expression. Transgenes that express *mVenus*-tagged wild-type *unc-6(+)* from ectopic, non-ventral sources were constructed. A *Pmyo-2::unc-6(+)* transgene was predicted to drive UNC-6 expression in the pharyngeal muscle in the anterior of the animal [30,31]; and a *Pslt-1::unc-6(+)* transgene was predicted to express UNC-6 in dorsal body wall muscle [32]. *Pmyo-2::unc-6* drove strong UNC-6::mVenus expression in the pharynx not observed in animals with *unc-6::gfp* driven from the endogenous *unc-6* promoter (Fig 6A). No detectable dorsal expression of UNC-6::mVenus from the *Pslt-1::unc-6(+)* was detected, possibly due to weaker expression from this transgene. Despite visible UNC-6::mVenus punctate expression on the pharynx in *Pmyo-2::unc-6*, no diffusible extracellular UNC-6::mVenus could be detected. This is similar to experiments using immunofluorescence to detect endogenous UNC-6 and UNC-6 from transgenes (expression on the cell of origin could be detected but diffusible UNC-6 could not be conclusively determined) [18,27].

*Pmyo-2::unc-6* or *Pslt-1::unc-6* transgenic animals in a wild-type *unc-6(+)* background caused no significant defects in the number of VD/DD axons protruding from the ventral cord or the number passing the lateral midline (Figs 6B–6D, 7A and 7B). *Pslt-1::unc-6* caused a weak but significant decrease in the number of axons reaching the dorsal cord, but *Pmyo-2::unc-6* did not. Thus, VD/DD axon guidance was largely unaffected by ectopic *unc-6* expression. The weak effect of *Pslt-1::unc-6* in reaching the dorsal cord could be due to VD growth cones making contact with dorsal muscles expressing *unc-6* and might represent a short-range interaction rather than a long-range effect. This suggests that ectopic expression of *unc-6(+)* from the *myo-2* and *slt-1* promoters did not strongly redirect VD/DD axon guidance.

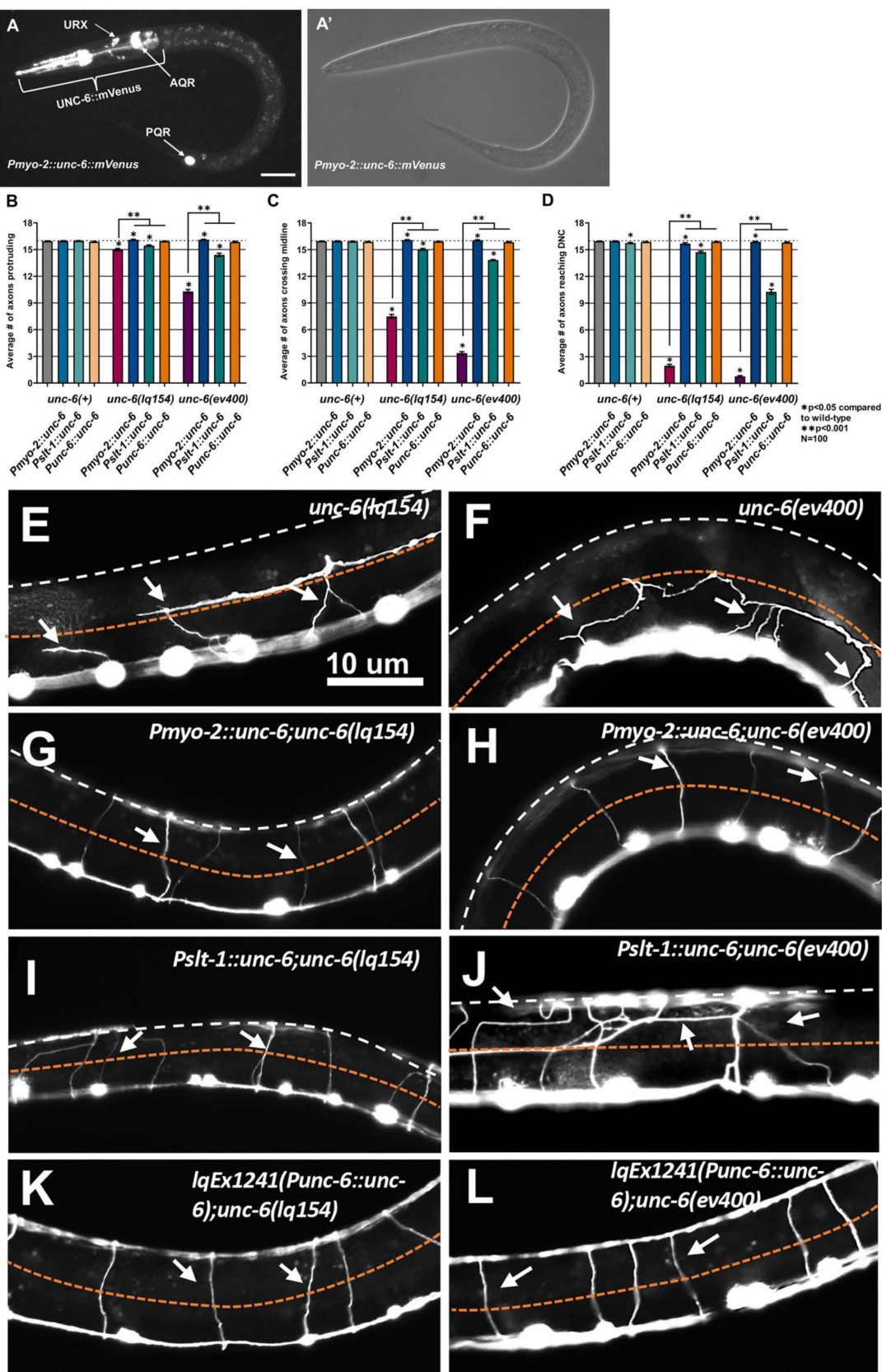

**Fig 6. _unc-6_ ectopic expression does not perturb VD/DD axon guidance.** A) Pharyngeal micrograph. A) A fluorescent micrograph of a late L1 larval animal with _Pmyo-2::unc-6::mVenus_ expression in the pharynx (bracket). The scale bar

represents 5 μm. *Pgcy-32*::yfp expression in the URX, AQR, and PQR neurons from the *Pgcy-32::yfp* co-transformation marker are indicated. A' is a DIC micrograph of the same animal. Anterior is to the left and dorsal is up. B–D) Graphs showing the average number of axons protruding from the ventral nerve cord, passing the lateral midline, or reaching the dorsal nerve cord, as described in Fig 2. Error bars represent standard error of the mean. Statistical significance was determined by a two-tailed *t*-test with unequal variance. E–L) Representative images of VD/DD axons in indicated genotypes. Approximate lateral midline is indicated by the dashed orange line and the dorsal nerve cord is indicated by the dashed white line. White arrows point to axons. Anterior is to the left and dorsal is up. The scale bar in E represents 10 μm for all images.

### Ectopic *unc-6* expression did not strongly affect VD/DD dorsal axon guidance

VD growth cone polarity below and above the lateral midline was also not significantly affected by ectopic *unc-6* expression from the *Pmyo-2::unc-6* and *Pslt-1::unc-6* trangenes. Expression from the endogenous *unc-6* promoter from the *Punc-6::unc-6* transgene also resulted in no defects in VD growth cone polarity (Fig 7). All transgenes significantly rescued VD growth cone polarity defects of *unc-6(lq154)* and *unc-6(ev400)* mutants (Fig 7).

Together, these data indicate that ectopic expression of *unc-6* did not perturb VD growth cone polarity or VD/DD dorsal axon guidance. It is possible that the transgenes do not produce amounts of diffusible UNC-6 necessary to affect axon guidance. However, as described below, AVM and PVM guidance was perturbed by ectopic UNC-6, indicating that diffusible UNC-6 was produced from these transgenes.

*Pmyo-2::unc-6* and *Pslt-1::unc-6* rescued VD/DD axon guidance and VD growth cone polarity in *unc-6(ev400)* and *unc-6(lq154)* mutants (Figs 6 and 7). While the nature of this rescue is not completely understood, it could be due to regulatory elements present in the introns of the *unc-6* gene that might drive *unc-6* expression in endogenous locations. Indeed, a *unc-6::mVenus* construct lacking a promoter and consisting only of the *unc-6::mVenus* coding region rescued the uncoordinated locomotion of *unc-6(ev400)*. This suggests that the *unc-6* coding region contains regulatory elements that can potentially drive endogenous *unc-6* expression. In any case, ectopic *unc-6::mVenus* expression did not affect VD/DD axon guidance or growth cone polarity, but did significantly perturb AVM/PVM axon guidance.

### Ectopic UNC-6 expression perturbed AVM and PVM axon guidance

*Pmyo-2::unc-6* and *Pslt-1::unc-6* rescued AVM and PVM ventral guidance defects in *unc-6* mutants, similar to rescue of VD/DD guidance described above (Fig 8A and 8B). However, in a wild-type background, *Pmyo-2::unc-6* and *Pslt-1::unc-6* caused weak but significant defects in AVM and PVM ventral axon guidance (8–22%) (Fig 8A and 8B). AVM and PVM axons extended anteriorly, similar to *unc-6* mutants. Unlike *unc-6* mutants, axons extended anteriorly only for a short distance before turning ventrally, eventually reaching the ventral nerve cord (Fig 8C and 8D). Expression from the endogenous *unc-6* promoter did not significantly perturb AVM/PVM ventral guidance (Fig 8). This suggests that ectopic expression of *unc-6* might interfere with the ventral axon guidance of AVM and PVM in a wild-type background, albeit weakly.

A concern with these ectopic expression studies is that transgenically-expressed UNC-6::mVenus might not be able to act at a distance, possibly not diffusing from the site of expression. The fact that AVM/PVM axon guidance was significantly disrupted by ectopic UNC-6::mVenus (Fig 8) suggests that transgenically expressed UNC-6::mVenus can act at a distance away from the site of expression.

## Materials and methods

### Genetic methods

Experiments were performed at 20 °C using standard *C. elegans* techniques [33]. Mutations used were *lqEx1241* [*Punc-6::Venus::unc-6*]; LGII: *juIs76* [*Punc-25::gfp*]; LGIV: *zdIs4*

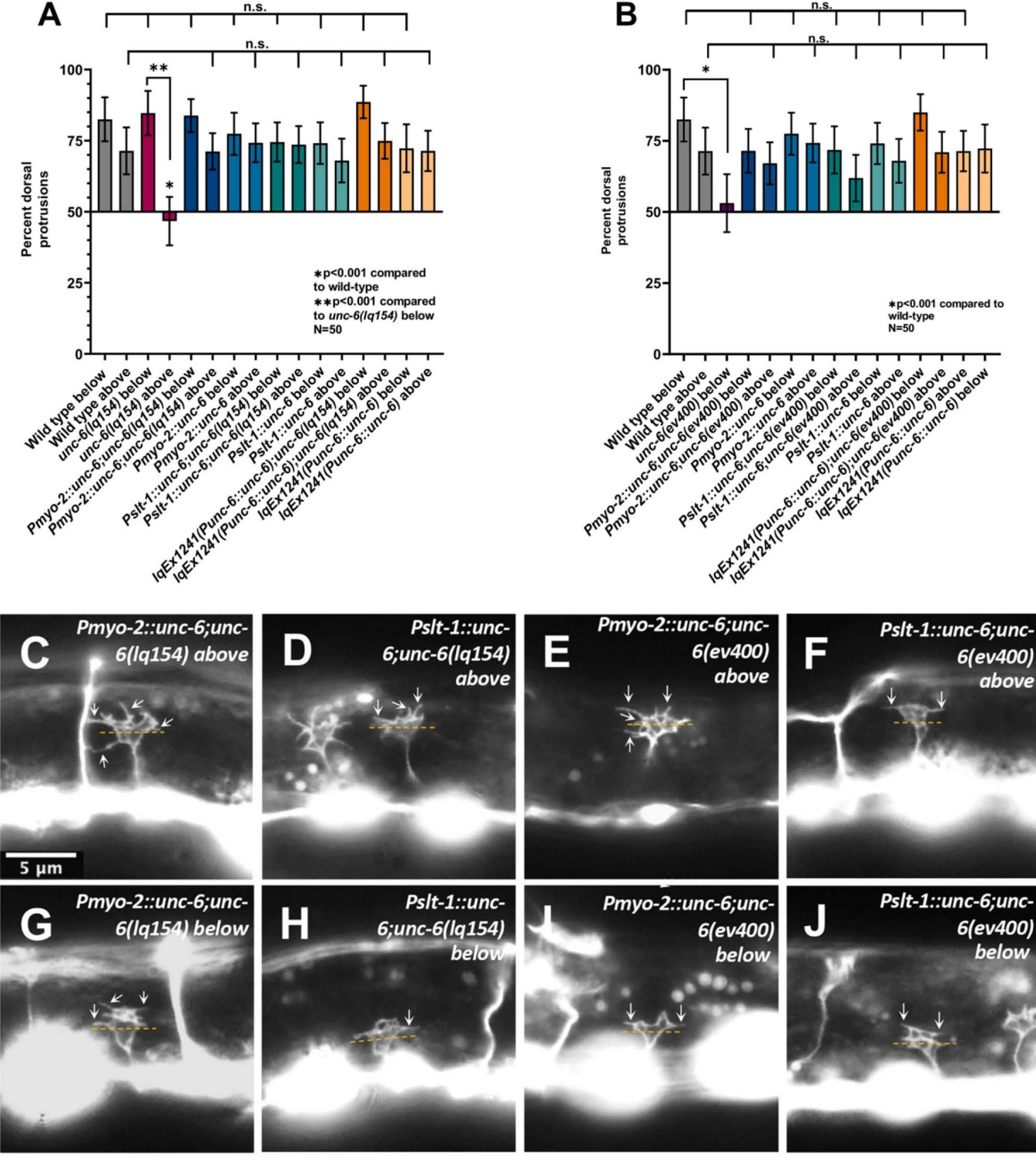

**Fig 7. *unc-6* ectopic expression does not perturb VD growth cone polarity.** A, B) Graphs of VD growth cone polarity as described in Fig 4. Error bars represent 2× the standard error of proportion. Statistical significance was determined using Fisher's exact test. Data for *wild-type* come from Fig 4A) *unc-6* transgenes in the *unc-6(lq154)* background. B) *unc-6* transgenes in the *unc-6(lq154)* background. B) C–J) Fluorescent micrographs of VD growth cones. Arrows point to filopodial protrusions. Approximate midline used to divide the growth cone into equal sections is indicated by dashed yellow line. The scale bar in C represents 5 μm for all images.

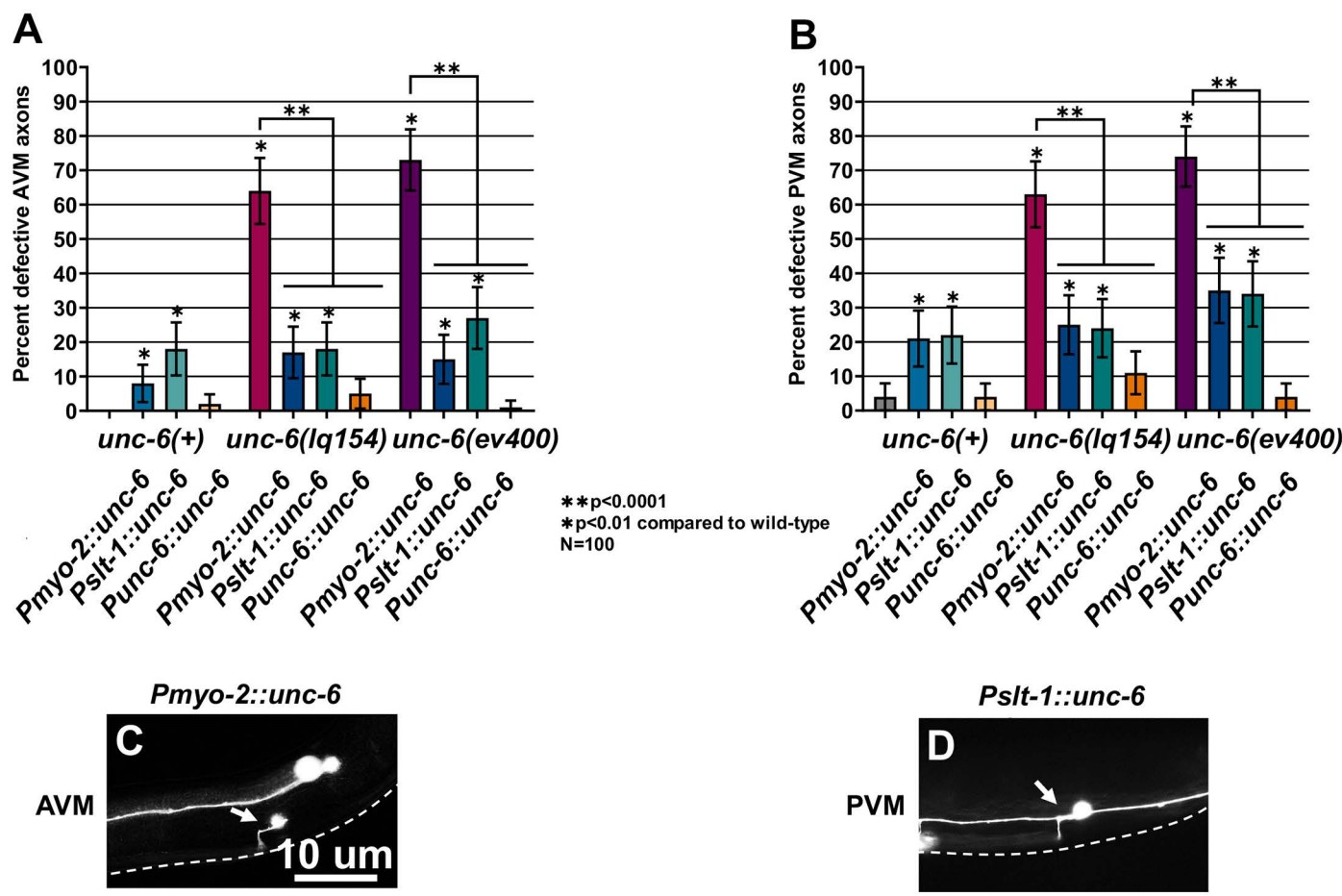

**Fig 8. *unc-6* ectopic expression perturbs AVM/PVM ventral axon guidance.** A, B) Graphs showing the percentage of defective AVM and PVM axons (failure to reach ventral nerve cord, branching, or turning at >45° angle during migration) as described in Fig 3. Error bars represent 2× the standard error of proportion. Statistical significance was determined by Fisher's exact test. C, D) Fluorescent micrographs of AVM and PVM in *wild-type* animals harboring *Pmyo-2::unc-6* (C) or *Pslt-1::unc-6* (D). Arrows point to the anteriorly-directed AVM or PVM axons. The ventral nerve cord is represented by a dashed line.

[*mec-4::gfp*]; LG V: *kyIs179 [Punc-86::gfp]*. LGX: *unc-6(ev400, lq154), lqIs170 [Prgef-1::vab-10ABD::gfp]*, *lqIs379 [Pmyo-2::Venus::unc-6]*, *lhIs11 [Pacr-2::mCherry]*. Chromosomal locations not determined: *lqIs401 [Pslt-1::Venus::unc-6]* and *lhIs6 [Punc-25::mCherry]*. Transgenic strains were obtained by microinjection of plasmid DNA into the germline. Multiple extrachromosomal transgenic lines of *Pmyo-1::Venus::unc-6* and *Pslt-1::Venus::unc-6* were established and integrated into the genome via standard techniques [34]. One integrated strain of each was chosen for further analysis. *lqEx1241 [Punc-6::Venus::unc-6]* was maintained as an extrachromosomal array. *lqEx1418-lqEx1420* transgenes contain the promoterless *unc-6::mVenus* coding region generated by inverse PCR of the *Pmyo-2::unc-6::mVenus* transgene to remove the promoter. *lqEx1421* and *lqEx1422* transgenes contained a restriction fragment of *Pmyo-2::unc-6* lacking most of the *myo-2* promoter. Mutations were confirmed using PCR genotyping and sequencing. Wormbase [35] was utilized for *C. elegans* informatics.

### Transgene construction

Transgenes were created using a *pVns::unc-6* plasmid containing the *unc-6* coding region as well as the upstream promoter and 3' UTR regions [27]. The *unc-6* promoter was replaced in

with the *myo*-2 and *slt-1* promoters. The promoterless *unc-6::mVenus* construct was generated by inverse PCR of the *Pmyo-2::unc-6::mVenus* transgene to remove the *myo-2* promoter. Each plasmid was confirmed using PCR and sequencing.

## CRISPR/Cas9 to generate *unc-6(lq154)*

CRISPR/Cas9 genome editing was used to insert a transmembrane signal sequence into the 3' end of *unc-6*. sgRNAs were engineered to target the 3' end of exon 12 and the 3' UTR. A donor plasmid containing left and right homology arms with recoded sgRNA regions, GFP, a synthetic intron flocked by LoxP sites that contains a hygromycin resistance cassette, a re-coded exon 13, and the coding region for the *cdh-3* transmembrane segment was used as a repair template. The repair plasmid sequence is the sequence of the genome edit (pnu1884; S1 Data). A mixture of sgRNAs, Cas9, and donor homology plasmid was injected into the gonads of N2 animals. Plates were treated with hygromycin and surviving animals were screened for the insert by PCR genotyping. The *unc-6(lq154)* genome edit was confirmed by sequencing the region from *unc-6(lq154)* genomic DNA. Genome editing reagents were provided by InVivo Biosystems (Eugene, OR, USA).

 sgRNA 1: ATTATGGATAAGgtaagaac
 sgRNA 2: agattggatcaggagtcaca
 *unc-6(lq154)* was outcrossed to N2 animals three times before analysis.

## RNA seq on *unc-6(lq154)*

Total RNA was isolated from mixed stage *unc-6(lq154)* animals (strain LE5443, *unc-6(lq154) X; juIs76 II*) as described previously. RNA seq libraries were constructed using the NEBNext stranded mRNA library kit. Sequencing was conducted on a Nextseq 2000 instrument with paired-end 150-bp sequencing. Preprocessing of FASTQ files was completed using fastp (0.23.2) [36], which included adapter trimming, per-read cutting by quality score, global trimming, and filtering out bad reads. HISAT2 (version 2.2.1) [37] was used to align reads to a *C. elegans* reference genome [release WBcel235, version WBPS14 (WS271)] that had been edited to include the *unc-6(lq154)* genome edit sequence. Resulting BAM files were analyzed in the Integrated Genome Viewer [38,39]. Raw reads can be found in the Sequence Read Archive PRJNA1093133.

## Imaging and quantification of axon guidance defects

The AVM and PVM axons were visualized with a *Pmec-4::gfp* transgene, *zdIs4,* which is expressed in the touch receptor neurons [40]. AVM and PVM axons were considered defective if they failed to reach the ventral nerve cord, wandered laterally at more than a 45° angle during ventral migration, or had ectopic processes. HSN axons were scored using the *Punc-86*::gfp transgene *kyIs179.* HSN axons were considered defective if they failed to extend ventrally to the ventral nerve cord. Axons were scored in L4 or pre-gravid adults. 100 animals were scored for each genotype and the percent defective axons was determined. Significance was determined by using a Fisher's exact test. VD/DD neurons were visualized using a *Punc-25::gfp* transgene, *juIs76,* which is expressed in all GABAergic neurons [41]. Of the 19 VD/DD axons, 18 extend commissures on the right side of the animal. The VD1 commissure on the left side was not scored. In wild-type animals, an average of 16 axons are observed due to fasciculation. Three reference points were used to quantify axon guidance defects: protrusion from the ventral nerve cord, crossing the lateral midline, and reaching the DNC. In mutants where less than 16 axons were observed only the observable axons were scored. In mutants where more than 16 axons were observed all observable axons were scored. Significance

between genotypes was determined using a two-tailed t-test with unequal variance. 100 animals were scored per genotype.

## Growth cone imaging and quantification

VD growth cones were visualized using a *Punc-25::gfp* transgene, *juIs76* [41]. Growth cones were imaged as previously described [4]. Animals were collected at approximately 16 hours post-hatching and placed on a 2% agarose pad with 5 mM sodium azide in M9 buffer. Mutants were harvested at different timepoints to account for differences in development time. Growth cones were imaged using a Qimaging Rolera mGi camera on a Leica DM5500 microscope and analyzed with ImageJ. To determine growth cone polarity, the growth cone was divided in half into dorsal and ventral subsections relative to the ventral nerve cord. Percent dorsal protrusions was determined by counting the filopodial on the dorsal half and dividing by the total filopodia. Per genotype, 50 growth cones were analyzed below the midline and above the midline. Statistical significance was determined using a Fisher's exact test.

## VAB-10ABD::GFP imaging

F-actin was analyzed as previously described [4]. The F-actin binding domain of VAB-10 fused to GFP [42] was used to visualize F-actin within the growth cone as described previously [4,16]. To control for growth cone size and shape the growth cone was visualized using a soluble mCherry marker. F-actin accumulation was analyzed using ImageJ. For each growth cone, five line scans were drawn from the dorsal edge to the ventral (see results). Pixel intensity was measured as a function of distance from the dorsal edge. Pixel intensity was averaged and normalized to the volumetric mCherry fluorescence in line scans from the dorsal and ventral half of the growth cone. This normalized ratio was determined for 50 growth cones below and above the midline in *unc-6(lq154)* and 50 in wild-type. Statistical significance was determined using a two-tailed t-test with unequal variance on the average normalized ratios.

## Discussion

Previous studies have shown that UNC-6/Netrin directs dorsal VD growth cone outgrowth by first polarizing the growth cone via the UNC-5 receptor such that filopodial protrusions and F-actin are biased dorsally on the growth cone. UNC-6/Netrin then maintains this polarity as the growth cone migrates dorsally, with the UNC-40 receptor driving protrusion dorsally, and the UNC-5 receptor inhibiting protrusion ventrally, resulting in net dorsal outgrowth [4,15,16,43,44]. The initial polarization of the growth cone by UNC-6/Netrin could involve a close-range interaction, whereas diffusible UNC-6/Netrin might be required for the longer-range maintenance of polarity. To test this model, an *unc-6(lq154)* genome edit was created that has the potential to encode a transmembrane version of UNC-6, which is not expected to diffuse and is expected to act only at short range.

### Short and long-range roles of UNC-6

*unc-6(tm)* behaved as a hypomorphic *unc-6* mutation, and was less severely uncoordinated and displayed fewer VD/DD commissural axon outgrowth defects compared to the *unc-6(null)*. VD growth cones near the ventral nerve cord (ventral to the lateral midline) were polarized similar to wild-type, with filopodial protrusions and F-actin biased dorsal-ward, whereas as *unc-6(null)* mutants were unpolarized at this point, and in fact very few extended dorsally past the lateral midline. VD growth cones further away from the ventral nerve cord, dorsal of the lateral midline, were unpolarized in *unc-6(tm)*. This suggests that *unc-6(tm)* growth cones were initially polarized normally, but lost polarity as they moved away from the

ventral nerve cord, explaining the hypomorphic VD/DD axon guidance defects of *unc-6(tm)* mutants.

At the time of VD growth cone outgrowth, UNC-6 is expressed in the VA and VB motor neurons that extend axons in the ventral nerve cord [18]. In the adult ventral nerve cord, VD and DD axons are often in contact with and dorsal to the VA and VB axons [45]. An intriguing hypothesis is that *unc-6(tm)* retains close-range functions of UNC-6, but lacks UNC-6 functions at longer range. RNA-seq shows that *unc-6(tm)* can encode a transmembrane version of UNC-6 (UNC-6(TM)). Possibly, the VD growth cone is initially polarized with a close-range, potentially contact-mediated interaction with UNC-6, but maintenance of polarity requires a diffusible, long-range role of UNC-6, and membrane anchored UNC-6(TM) cannot provide this long-range function. It is also possible that overall levels of UNC-6 are reduced in *unc-6(tm)*, resulting in this hypomorphic phenotype. However, these results are consistent with *unc-6(tm)* encoding a transmembrane UNC-6 molecule that cannot diffuse at long range.

Consistent with this idea, *unc-6(tm)* and *unc-6(null)* mutants displayed similar levels of AVM and PVM ventral axon guidance defects. AVM and PVM cell bodies reside laterally away from the ventral nerve cord, and thus require longer-range UNC-6, presuming the source is the VA and VB neurons in the ventral nerve cord. This suggests that *unc-6(tm)* is deficient for longer-range roles of UNC-6. Together, these data indicate that a close-range interaction of the VD and DD axons with the *unc-6*-expressing VA and VB motor axons polarizes the growth cone, and that diffusible UNC-6 is required to maintain this polarity as the growth cones move dorsally away from the ventral nerve cord. Thus, UNC-6 might have distinct close-range and long-range guidance functions. A similar close-range interaction might occur in the vertebrate spinal cord, where commissural axon outgrowth requires haptotactic interactions of the axons with the ventricular zone ependymal stem cell processes along the outgrowth path that express Netrin1, and not the Netrin1 source in the floorplate [23–26]. Furthermore, *Drosophila* Netrin has short-range effects independent of DCC [22]. This is consistent with studies in *C. elegans* showing that UNC-5 but not UNC-40/DCC is required for initial VD growth cone polarity [4,46], and that UNC-40 drives growth cone protrusion in response to UNC-6 [4,15,16]. In the Polarity/Protrusion model, initial and ongoing VD growth cone polarity relies on the UNC-5 receptor, which also inhibits growth cone protrusion laterally and ventrally. UNC-40/DCC stimulates protrusion dorsally in response to UNC-6, which results in net dorsal growth cone migration. Growth cone size in *unc-6* mutants is not increased as it is in *unc-5* mutants, possible reflecting this dual role of UNC-6 in both stimulating and inhibiting protrusion.

While *unc-6(lq154)* is predicted to encode a transmembrane anchored version of UNC-6 as evidenced by transcripts that encode such a molecule (Fig 1), it is also possible non-membrane-anchored, diffusible UNC-6 is also produced, possibly from the transcripts that fail to remove the introns upstream of the transmembrane domain coding region. If diffusible UNC-6 is produced in *unc-6(lq154)*, it is possible that it is not produced in sufficient quantity to mediate the long-range effects of UNC-6 (Fig 9).

## Ectopic UNC-6 expression did not perturb VD/DD axon guidance or VD growth cone polarity

The data presented here, along with previously reported *unc-6* expression in the VA and VB neurons [18], suggest that a ventral directional UNC-6 signal might polarize the VD growth cone at short range, and that diffusible UNC-6 is required to maintain polarity. If the role of long-range UNC-6 is solely to maintain growth VD cone polarity established earlier by short range interactions, then ectopic expression of UNC-6 from non-ventral sources should not perturb VD axon guidance or growth cone polarity.

## A. Short-range UNC-6

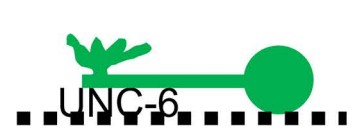

**Establish VD growth cone polarity**

## B. Long-range UNC-6

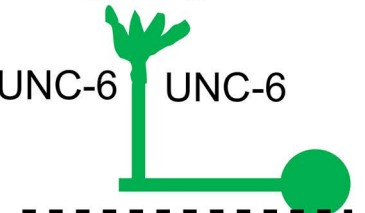

**Maintain VD growth cone polarity and protrusion**

## C. Long-range UNC-6

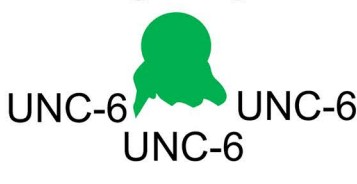

**Direct AVM/PVM ventral axon guidance**

**Fig 9. A model of short- and long-range roles of UNC-6.** A) UNC-6 from the ventral nerve cord (dashed line) acts at short range to polarize the VD growth cone dorsally. Once established by this short-range interaction, this polarity cannot be redirected by ectopic expression of UNC-6. B) Diffusible UNC-6 might mediate a long-range role of UNC-6 to maintain VD growth cone polarity and to regulate protrusion with a dorsal bias. UNC-40 activity stimulates protrusion dorsally and UNC-5 inhibits protrusion ventrally. C) Long-range, diffusible UNC-6 polarizes AVM and PVM ventral axon outgrowth. Ectopic UNC-6 can perturb this process, possibly because the polarity event is established by diffusible UNC-6 over long range, as opposed to VD growth cone polarity which is a close-range interaction.

Previous studies showed that ectopic *unc-6* expression could redirect or perturb axon guidance, including dorsal-ventral placement of the sublateral nerves and SDQL migration [47,48]. In studies reported here, ectopic expression of UNC-6::mVenus from *myo-2* promoter in the pharynx or from the *slt-1* promoter in the dorsal muscle did not interfere with VD/DD axon guidance or VD growth cone polarity. This is consistent with the idea that a short-range interaction with UNC-6 in the ventral cord is required to polarize that VD growth cone, and that long-range UNC-6 cannot re-direct this polarity once established. Together with previous results, this suggests that the role of long-range UNC-6 is to maintain growth cone polarity established by the short-range interaction, as well as to control growth cone protrusion, inhibiting protrusion ventrally via the UNC-5 receptor and stimulating protrusion dorsally via the UNC-40 receptor [4,15,16,43,44].

*Pslt-1::unc-6* did cause weak defects in the ability of VD/DD axons to complete the final stage of their trajectory of reaching the dorsal nerve cord. All axons migrated dorsally past the lateral midline, but some failed to reach the dorsal nerve cord. Possibly, VD/DD growth cone directly encounter UNC-6 expression on the dorsal muscles from *Pslt-1::unc-6*. Indeed, VD growth cones undergo a complex interaction with the dorsal muscles in order to grow past the very robust muscle-to-hypodermis-to-cuticle adhesion [29]. This might represent a short-range role of UNC-6. It is possible that interaction with UNC-6 at short range or through contact is able to redirect growth cones, whereas long-range interactions cannot.

### Ectopic UNC-6 expression perturbed AVM/PVM axon guidance

Ectopic expression of UNC-6::mVenus significantly perturbed AVM/PVM ventral axon guidance. In a wild-type background, ectopic UNC-6::mVenus expression caused failure of initial ventral growth of the AVM and PVM axon. Similarly, previous results showed that sublateral nerve placement and SDQL migration were affected by ectopic UNC-6 [47,48]. This indicates that different axon guidance events respond differently to ectopic UNC-6. In the case of the VD/DD growth cones, initial polarity is established in the ventral nerve cord near the site of UNC-6 expression. In the case of sublateral nerves, SDQL, and AVM/PVM, the axon guidance

events occur at a distance from the ventral site of UNC-6 expression and requires diffusible UNC-6. Possibly, long-range polarity events that require diffusible UNC-6 can be perturbed by ectopic UNC-6, whereas short-range interactions cannot. This could be due to different environments involved in UNC-6 function in each case, such as interactions with distinct extracellular molecules at short and long range. It could also be due to direct contact of VD/DD axons with UNC-6 expressing axons in the ventral cord, such as the vAs and VBs. Indeed, in serial electron microscopic reconstructions of the ventral nerve cord, the VD and DD axons are often in contact with the VA and VB axons and are often dorsal to them [45].

Results presented here are consistent with the Polarity/Protrusion model of dorsal growth cone migration away from UNC-6, wherein UNC-6/Netrin first polarizes the VD growth cone in a short-range or contact-mediated event, and then maintains that polarity at a longer range as the growth cone migrates away from the ventral nerve cord (Fig 9). Ectopic expression of UNC-6/Netrin did not severely perturb dorsal VD/DD axon guidance, which would be expected in a chemotactic gradient model. The gradient model implies that growth cones dynamically sense Netrin levels and change their behavior accordingly. For example, growth cones attracted to Netrin would protrude more robustly in the direction of Netrin, and repelled growth cones the opposite. This is not what was observed in the VD/DD neurons by ectopic expression of UNC-6.

The SOAL model of ventral axon growth in response to UNC-6 postulates that long-range UNC-6 polarizes UNC-40 to the ventral region of the HSN neuron, which then drives axon protrusion ventrally. A similar mechanism could be occurring in the AVM/PVM neurons. Results presented here indicate that ectopic UNC-6 expression can perturb AVM/PVM axon guidance, suggesting that a polarity event that relies on long-range UNC-6 can be affected by ectopic UNC-6. This again suggests that short-range and long-range UNC-6 might act by distinct mechanisms. In either case (dorsal guidance (the Polarity/Protrusion model) and ventral guidance (the SOAL model)), the role of UNC-6 appears to be to polarize the pro-protrusive activity of the UNC-40/DCC receptor in the direction of axon outgrowth.

## Supporting information

**S1 Data. This is a Genbank file containing the annotated sequence of the repair template plasmid used to create *unc-6(lq154)* by Cas9 fgenome editing, called pNU1884.** (GB)

**S2 Data. This is a compressed ZIP archive containing a FASTA *C. elegans* reference X chromosome edited to include the *unc-6(lq154)* genome edit and a GTF file of the *unc-6(lq154)* edited region.** (ZIP)

**S3 Data. This is a compressed ZIP archive of the data and analysis on which graphs in the figures are based.** The dataset for each figure is indicated. (ZIP)

## Acknowledgments

Thanks to the Lundquist and Ackley labs for discussion and suggestions, the National Institutes of Health Kansas Infrastructure Network of Biomedical Research Excellence (P20GM103418), the KU Center for Genomics, and Wormbase. Some *C. elegans* strains were provided by the *Caenorhabditis* Genetics Center, funded by National Institutes of Health (P40 OD010440). High-throughput sequencing was conducted by the KU Genome Sequencing Core, part of the National Institutes of Health *Center for Molecular Analysis of Disease Pathways* (P30GM145499).

## Author contributions

**Conceptualization:** Kelsey M. Hooper, Erik A. Lundquist.

**Data curation:** Kelsey M. Hooper.

**Formal analysis:** Kelsey M. Hooper.

**Funding acquisition:** Erik A. Lundquist.

**Investigation:** Kelsey M. Hooper, Vedant D. Jain, Celeste J. Gormly, Brian J. Sanderson.

**Project administration:** Erik A. Lundquist.

**Visualization:** Kelsey M. Hooper.

**Writing – original draft:** Kelsey M. Hooper.

**Writing – review & editing:** Erik A. Lundquist.

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
