## [Decision Letter · Decision Letter 0]

21 Jun 2024

Dear Dr Lundquist,

Thank you very much for submitting your Research Article entitled 'Short- and long-range roles of UNC-6/Netrin in dorsal-ventral axon guidance in vivo in Caenorhabditis elegans' to PLOS Genetics.

The manuscript was fully evaluated at the editorial level and by independent peer reviewers. The reviewers appreciated the attention to an important problem, but raised some substantial concerns about the current manuscript. Based on the reviews, we will not be able to accept this version of the manuscript, but we would be willing to review a much-revised version. We cannot, of course, promise publication at that time.

Should you decide to revise the manuscript for further consideration here, your revisions should address the specific points made by each reviewer. As you will see, additional experimental work will likely be required to address the points raised in review, especially regarding the expression and localization of the putatively membrane localized unc-6. We will also require a detailed list of your responses to the review comments and a description of the changes you have made in the manuscript.

If you decide to revise the manuscript for further consideration at PLOS Genetics, please aim to resubmit within the next 60 days, unless it will take extra time to address the concerns of the reviewers, in which case we would appreciate an expected resubmission date by email to plosgenetics@plos.org.

If present, accompanying reviewer attachments are included with this email; please notify the journal office if any appear to be missing. They will also be available for download from the link below. You can use this link to log into the system when you are ready to submit a revised version, having first consulted our Submission Checklist .

PLOS has incorporated Similarity Check , powered by iThenticate, into its journal-wide submission system in order to screen submitted content for originality before publication. Each PLOS journal undertakes screening on a proportion of submitted articles. You will be contacted if needed following the screening process.

We are sorry that we cannot be more positive about your manuscript at this stage. Please do not hesitate to contact us if you have any concerns or questions.

Yours sincerely,

Andrew D. Chisholm

Academic Editor

PLOS Genetics

Giovanni Bosco

Section Editor

PLOS Genetics

Reviewer's Responses to Questions

**Comments to the Authors:**

Reviewer #1: The manuscript reports original, important, exciting and potentially highly significant findings for developmental neurobiologists and geneticists. A dual role for UNC-6/Netrin in dorsal-ventral axon guidance is claimed. The first role involves a close-range interaction that polarizes the growth cone, while the second role, which requires diffusible UNC-6/Netrin, acts in the longer-range for maintenance of growth cone polarity. Contrary to earlier assumptions, the manuscript provides a significant new finding: ventrally derived UNC-6 is not essential for dorsal-ventral axon guidance, suggesting that its role is permissive rather than instructive. These findings significantly contribute to our understanding of the mechanisms underlying axon guidance, with potential broad implications for developmental biology and neurobiology. However, key pieces of evidence are required to support the authors claims, as central conclusions are, as it stands, unsupported by data.

Major concerns

1. There is no evidence for UNC-6(TM) being at the plasma membrane, or at least not being secreted. The membrane-anchored transmembrane UNC-6 generated by genome editing is an elegant and clever strategy employed by the authors to explore potential short- and long-range roles of UNC-6/Netrin in dorsal-ventral axon guidance, and serves as the foundation for the main claims of the manuscript. However, no evidence is provided that the protein UNC-6(TM) is indeed present at the plasma membrane, which given its significance, is key to support the interpretations. Possible experiments for this:

1.1- Is GFP-tagged UNC-6(TM) visible at the membrane level of by fluorescence microscopy? Likely endogenous levels are too low; if so, mention it explicitly in text. Could a multicopy transgene similar to the endogenous UNC-6(TM) be made with the brightest fluorescent protein possible (e.g., mNG) to show that it is at the membrane?

Alternatively, use heterologous cells to visualize UNC-6(TM) (e.g. coelomocytes or intestinal cells in the worm, or express a corresponding construct in mammalian or Drosophila cells? Of course there are more difficult and time-consuming alternatives such as FRET or split-GFP with another transmembrane protein expected to be close to UNC-6, such as UNC-40, but that may be out of the scope of the manuscript.

1.2- At the very least, rescue experiments of unc-6(ev400) and unc-6(lq154) mutants using the unc-6(tm) transgene should be done to demonstrate that membrane-bound UNC-6(TM) is not able to rescue when it is expressed distantly, e.g., from the pharynx or from dorsal muscles only, as it is expected to be membrane-bound and not reach the developing AVM, PVM, HSN or VD/DD. For generating such transgene, for instance, the cassette obtained by genome editing could be PCR amplified from the CRISPR strain- about 12 kb, or obtained by cloning a minigene with the corresponding TM.

2. An alternative interpretation for why Pslt-1::unc-6 or Pmyo-2::unc-6 transgenes are sufficient to rescue distant neuronal developmental processes could be that exons of the unc-6 cDNA contain regulatory/promoter regions of the gene unc-6, thus driving natural endogenous expression in developing cells. This possibility undermines a central conclusion of the manuscript that " ectopic unc-6(+) expression from non-ventral sources could rescue dorsal and ventral guidance defects in unc-6(tm) and unc-6(null). Thus, a ventral directional source of UNC-6 was not required for dorsal-ventral axon guidance, and UNC-6 can act as a permissive, not instructive, cue for dorsal-ventral axon guidance".

A promotor-less control should be done, where the unc-6 cDNA is not driven by any promoter and used to test for rescue. Fortunately, the limited rescue of AVM and PVM is somewhat reassuring, but different neurons may use distinct regulatory signal for unc-6 gene expression. A promoter-less construct would nail it and increase the significance of the findings.

3. Line 434: “UNC-6 might have distinct close-range and long-range guidance functions”. Are these functions distinct or could it be that the close and long-range UNC-6 guidance functions are the same, but that UNC-6(TM) suppresses UNC-6 long-range guidance functions for neurons that are too far to be affected by it?

As a related point, do UNC-6(TM) effects corelate with the presence of UNC-40 only or UNC-5/UNC-40 combination in the neurons under study?

4. Fig.4: Could the unc-6(tm) transgene (and not unc-6(+)) be expressed only from the dorsal aspect of the worm (Pslt-1::unc-6(TM)) to rescue the loss of polarity (protrusion + actin) in VD growth cones after the lateral midline in unc-6(lq154)?

5. Line 431/432: “These data indicate that a close-range interaction of the VD and DD axons with the unc-6-expressing VA and VB motor axons polarizes the growth cone”. This is a very appealing notion, but it is an overstatement as no evidence is provided for a direct relationship of unc-6 being expressed precisely from VA and VB motor axons and the polarization of VD and DD growth cones. The text should at least be toned down (ex: These data are consistent with a model where...)

This is also a concern at Line 218/219. About VA and VB neurons “which could be the source of UNC-6 that directs the VD growth cones dorsally”. Could the unc-6(tm) transgene be expressed from VA/VB specifically (is there a specific promoter?) to test for rescue of the loss of polarity after the lateral midline crossing of the VD growth cones in unc-6(TM) mutants.

6. Line 269: authors say “Pmyo-2::unc-6 and Pslt-1::unc-6 strongly rescued the HSN axon guidance defects of both mutants (Figure 9A)”, however no info about Pslt-1::unc-6 in figure 9A.

Minor comments

7. Has the CRISPR strain outcrossed? Not found in the M&M.

8. Fig 9: Can unc-6(+) from its own promoter (Punc-6) rescue the defects in HSN axons as well? And if so, does it induce defects in the WT?

- Does Pmyo-2::unc-6(+) still allow UNC-40 asymmetric localisation in HSN neurons?

9. About the defects of unc-6(lq154 UNC-6(TM)) allele. The same UNC-6(TM) transgene suggested in point 1.2 could be used for tissue-specific expression to see where it is needed to rescue unc-6(ev400) null neuronal development.

10. About the ectopic expression of UNC-6(+) -> what if there is a non-asymmetrical interactors of UNC-6 in the ECM or at the neuronal or cell surface that patterns the UNC-6 signal in a specific manner and this, even if the UNC-6/netrin source is ectopic/nondirectional? Scenario "C" for Fig 10 and Discussion.

11. From the RNA seq data, Introns surrounding the linker exon are sometimes integrated resulting in an early stop codon. Thus, there are potential copies of UNC-6 that are not transmembrane. Line 145/146 states that these transcripts probably enter the decay process because they do observe defects compared to WT. However, what if these copies were at very low level, or that the presence of GFP and the supplementary introns affected the folding of UNC-6 thus giving a partial loss of function? This is partially addressed in the Discussion, but more could be explained.

12. Only 1 line tested for Fig 8?

Finally, the figures are clear and the text is very well written and pleasant to read, just few very minor points noted:

- The word "Results" is missing (page 6, 117-118).

- Line 153-154 "The unc-6(ev400) null mutants are severely uncoordinated and display severe defects in the dorsal to ventral guidance of the VD and DD GABA-ergic motor axons." Should say ventral to dorsal.

- Figure 2, Line 592: The X-axis represents genotype and the Y axis the number "if" commissures observed in animals. Replace with OF

- Provide explanation of the meaning of the "N" in the VD figures. Figure 2. N=100 animals; Figure 4. N=50 axons; Figure 5. N=15 animals.

-Line 224. Replace "source cells" for "progenitor/ancestor cells", as source is used two other time to refer to netrin source in the same sentence.

Reviewer #2: In this manuscript the authors explore short-range functions of the axon guidance cue Netrin/UNC-6 in C. elegans. UNC-6 is a secreted protein essential for dorso-ventral navigation of axons. The original studies on UNC-6 led to a model, where UNC-6 forms a gradient with the highest concentration on the ventral side and is sensed by axons that either will grow towards or away from the highest concentration. Newer data in vertebrates and invertebrates have challenged this model and suggested distinctive short- and long-range roles for Netrin/UNC-6. Experiments presented in this paper are designed to further explore mainly the short-range function of UNC-6. The authors generated a membrane-tethered version of UNC-6, UNC-6(TM), that is not able to diffuse away from the source of production and should not have any long-range effects. The authors observed that UNC-6(TM) retained some (short-range) function specifically for dorsal migration supporting the idea that short- and long-range effects of UNC-6 are distinct. The authors show that this effect is due to an initial correct polarization of the growth cones that is lost as the growth cones move away from the source of UNC-6(TM). Finally, the authors show that ectopic expression of UNC-6 from different (non-ventral) sources can rescue defects of unc-6 mutants. This suggests that UNC-6 might act as a permissive rather than instructive cue and leads to a new model of UNC-6 action during axon navigation.

Together, the the experiments provide novel (and somewhat unexpected) insights into the role(s) of Netrin/UNC-6 in axon guidance, which seem to be more complex and rather different compared to the original model. As UNC-6 is one of the major evolutionary conserved axon guidance cues, the results presented here should be of interest to a wider audience. The experiments are well designed, carefully executed and clearly presented. The following issues should be addressed before publication.

The UNC-6(TM) construct

To generate a membrane bound version of UNC-6 the authors used CRISPR/Cas9 to introduce a new exon encoding a transmembrane (TM) domain at the C-terminus of UNC-6. For screening purposes, the CRISPR construct contained a selection cassette, which is designed to be removed by a Cre recombination event. Unfortunately, the Cre recombination failed, and the selection cassette could not be removed. The authors used RNAseq to analyze the resulting mRNA(s) produced from the modified unc-6(TM) locus. The results revealed that many transcripts contained a novel exon ('linker exon') encoding an additional 50 amino acids near the C-terminus. Presumably these additional amino acids do not impact the function of UNC-6, a potential if minor concern. More importantly, the authors found that the introns flanking the new 'linker exons' were not always spliced out, resulting in premature stop codons. These stop codons would truncate the modified UNC-6 protein before the TM domain resulting in a secreted and probably functional UNC-6 protein. The authors argue that the premature stop codon would likely lead to nonsense-mediated decay, but the fact that messages containing these introns are detectable at a significant frequency suggests that at least some secreted UNC-6 is produced. This obviously affects the interpretation of the results and needs to be discussed in some detail. The fact that the unc-6(tm) strain shows a partial lof phenotype clearly suggests that not enough 'wildtype' secreted unc-6 is produced. However, the authors use this as an argument that no functional secreted UNC-6 is produced (line 145: "This is evidenced by the unc-6(lq154) Unc and axon guidance phenotypes, which would not be expected if full-length UNC-6 without the transmembrane domain was being produced in unc-6(lq154).". This is an overstatement. The overall conclusions of the manuscript are still valid, but the interpretation and discussion of the results needs to consider the possibility that some secreted UNC-6 is present. In fact, this would provide an explanation for the observation that the effects of UNC-6(TM) are not strictly limited to the growth cones that are in immediate contact with the UNC-6(TM) producing cells. The growth cones can maintain their correct polarization for a while even after they have grown away from the ventral nerve cord and only lose their polarization after crossing the halfway point towards the dorsal side.

Ectopic expression of UNC-6

The authors found that expression of UNC-6 from a dorsal (rather than ventral) source can completely rescue unc-6 defects. Even more surprisingly, expression from the pharynx, which is even further away from the normal source of UNC-6 expression can also rescue - even defects in HSN axon navigation, which happens postembryoncially at a substantial distance from the pharynx. The authors correctly conclude that a ventral source of UNC-6 is not essential. However, the authors seem to imply (in their model) that ectopic expression results in uniform distribution of UNC-6 and that UNC-6 would diffuse freely and be mislocalized when ectopically expressed. This ignores the possibility that UNC-6 might be stably embedded in the basement membrane and not form a diffusion gradient. This seems likely since UNC-6 acts as a dorso-ventral guidance cue even during postembryonic development long after peak expression of unc-6 during embryogenesis. Depending on the mechanism of UNC-6 localization, ectopically expressed UNC-6 might even localize normally. For secreted proteins it is not possible to infer the final localization of the protein exclusively from the source of expression. In the discussion the authors need to be more explicit in their assumptions ('we assume UNC-6 is uniformly distributed' or whatever the assumption is) and they need to discuss alternative possibilities such as the one I laid out above.

Minor issues

The authors refer to the original paper describing unc-6 expression (Wadsworth et al. 1996) to argue where UNC-6 acts when membrane-bound. However, the UNC-6(TM) construct has a GFP tag, which should make it possible to directly observe which cells express UNC-6(TM)::GFP. The ability to detect the modified version of UNC-6 would also resolve the question of whether there is some secreted UNC-6. I assume the UNC-6(TM)::GFP is not detectably since the authors do not comment on it. The authors should explicitly say whether they attempted to detect the UNC-6(TM)::GFP and what the results were.

Figure 6D

Pslt-1::unc-6 completely rescues iq154 but only partially rescues ev400 (the null allele), whereas Pmyo-2::unc-6 completely rescues both iq154 and ev400. This is unexpected, do the authors have an explanation?

Figure 1B: The y-axis should be labelled (probably RPKM).

Figure 7: Are wt data copied from Figure 4? If so, this should be stated explicitly.

Reviewer #3: In recent years there has been an explosion of new studies challenging the paradigm of netrin function, suggesting that long range midline netrin guidance is dispensable, and that rather Netrin1 expression on the radial glial ependymal cells or ventricular guide neurons locally through a contact mediated process (DOMINICI et al. 2017; VARADARAJAN AND BUTLER 2017; YAMAUCHI et al. 2017; MORALES 2018). In this paper, Hooper and Lundquist take this question into the the c. elegans model by disrupting the potential availability of soluble netrin via genome editing to make potentially membrane anchored netrin (Unc-6). Using this they conclude that the membrane attached netrin does not rescue all phenotypes of the unc-6 loss of function mutants, suggesting a role for soluble netrin-1. Using overexpression of unc-6 from cells not restricted to the ventral portion of the worm, they do rescue axon guidance phenotypes, suggesting that the non-membrane bound netrin is not providing a directional cue, but rather a permissive cue. However, there are several pieces of data that are missing to completely support the authors conclusiosn. For examples, authors do not demonstrate whether netrin in membrane inserted and functions properly. Though authors mention that it is predicted to be a membrane protein by TMHMM, it is still unclear how they did it. Adding an explanation or experimentally addressing this concern will improve this manuscript significantly. Second, they do not demonstrate the localization of unc-6 when overexpressed. They want to conclude that the long range netrin is not directional but just permissive. However, that does not fit with the observation of axon defects upon overexpression of unc6 in a wildtype background. In addition to these two major concerns, several other suggestions are included below which would strengthen the study as well as make it more available to a broader readership.

1. For readers not steeped in c.elegans anatomy, a schematic of unc6, unc5, and unc40 in the different scenarios described at the beginning of the introduction would be incredibly helpful. Perhaps 2 schematics of the two different models of netrin guidance. The SOAL model is hard to imagine without this.

2. Figure 1D needs an x-axis label.

3. Scale bars are illegible in many places. Description of how Figure 2b/2c show that the animals are more or less uncoordinated is only in the figure legend, not briefly mentioned in results, which would be helpful to include.

4. Line 118 onwards: result section starts here, but isn’t labeled

5. Line 120: the potential to encode a transmembrane UNC-6 molecule:

Has it been tested? It is unclear why authors did not include a tag on the Unc6 to allow visualization or purification from membrane fraction, this would greatly strengthen their conclusions.

6. Figure 2GHI: authors are recommended to show more than mean/SEM, but show data points for each animal measured. This similar comment stands for all subsequent bar graphs in figures 3, 4, 5, 6.

7. I am confused why the phenotypes of the AVM and PVM axons not extended ventrally are not interpreted as a long range guidance, but permissive is what the abstract states.even the authors state many potential mechanisms that may be at play and have not yet been distinguished, but then claim one thing in the abstract, that may not jive with all the data in the paper, particularly the overexpression phenotypes. Couldn’t it be the shortrange does the initial polarization, and long range maintains polarization? How is this not guidance? Without showing the localization of overexpressed unc6 in the mutant background, this seems like a big jump in conclusions. If this ectopic expression is the demonstrative data for how unc6 functions, its is imperative to know where the unc6 is? How does ectopic expression of netrin disrupt ventral guidance, yet rescue ventral guidance in an unc6 null?

8. Line 232-234: authors mention: In a wild-type unc-6(+) background, Pmyo-2::unc-6 and Pslt-1::unc-6 expression caused no significant defects in VD/DD axon guidance and had no significant effect on VD growth cone polarity.

i. But the avg no of axons are decreased slightly in fig D (pstl1-unc6). Please explain what does that mean.

ii. Also, it would be prefered to show representative images first and then graphs in this figure.

9. Figure 4D: does that line bisect the growth cone, it looks like >> ½ is dorsal to the line.

10. Figure 8: Please explain figure details (A, B etc) in description properly. Why there are not representative images for wild type in figure 8.

**Have all data underlying the figures and results presented in the manuscript been provided?**

Reviewer #1: Yes

Reviewer #2: Yes

Reviewer #3: Yes

PLOS authors have the option to publish the peer review history of their article (what does this mean? ). If published, this will include your full peer review and any attached files.

**Do you want your identity to be public for this peer review?** For information about this choice, including consent withdrawal, please see our Privacy Policy .

Reviewer #1: No

Reviewer #2: No

Reviewer #3: No

---

## [Decision Letter · Decision Letter 1]

4 Dec 2024

Dear Dr Lundquist,

We are pleased to inform you that your manuscript entitled "Short- and long-range roles of UNC-6/Netrin in dorsal-ventral axon guidance in vivo in Caenorhabditis elegans" has been editorially accepted for publication in PLOS Genetics. Congratulations!

Yours sincerely,

Andrew D. Chisholm

Academic Editor

PLOS Genetics

Giovanni Bosco

Section Editor

PLOS Genetics

Aimée Dudley

Editor-in-Chief

PLOS Genetics

Anne Goriely

Editor-in-Chief

PLOS Genetics

Comments from the reviewers (if applicable):

Reviewer's Responses to Questions

**Comments to the Authors:**

Reviewer #2: This is the revised version of a previous manuscript. I'm not going to summarize the content. The authors have addressed all my concerns. The inability to visualize the extracellular tagged UNC-6 protein limits the ability to draw unambiguous conclusions especially for the ectopic expression experiments. Since nobody has ever been able to visualize extracellular UNC-6 there is nothing the authors can do about it. They addressed this problem by discussing in detail alternative interpretations of their results and softened their claims accordingly. Their main conclusion "that there are long and short roles of UNC-6" is supported by the results. The results also support their "Polarity/Protrusion model".

Regarding the promoterless unc-6 construct. The authors write " We conducted the promoterless control as suggested and found that this also rescued unc-6(ev400). Thus, the reviewers were correct that the unc-6 coding region likely contains regulatory regions. "

My understanding is the relevant construct is a plasmid. It seems much more likely that cryptic enhancer elements are in the vector backbone and not in the unc-6 coding part of the plasmid. The authors should maybe rephrase the above sentence to include this possibility.

Reviewer #3: The authors have addressed most of my concerns. I recognize that they prefer to show only mean and sem in their figures, but I do think that this is a mistake, and data transparency and scientific fields are largely moving away from this.

**Have all data underlying the figures and results presented in the manuscript been provided?**

Reviewer #2: Yes

Reviewer #3: None

PLOS authors have the option to publish the peer review history of their article (what does this mean? ). If published, this will include your full peer review and any attached files.

**Do you want your identity to be public for this peer review?** For information about this choice, including consent withdrawal, please see our Privacy Policy .

Reviewer #2: No

Reviewer #3: No

**Data Deposition**

http://datadryad.org/submit?journalID=pgenetics&manu=PGENETICS-D-24-00530R1

More information about depositing data in Dryad is available at http://www.datadryad.org/depositing . If you experience any difficulties in submitting your data, please contact help@datadryad.org for support.

**Press Queries**

---

## [Editor Report · Acceptance letter]

PGENETICS-D-24-00530R1

Short- and long-range roles of UNC-6/Netrin in dorsal-ventral axon guidance in vivo in Caenorhabditis elegans

Dear Dr Lundquist,

We are pleased to inform you that your manuscript entitled "Short- and long-range roles of UNC-6/Netrin in dorsal-ventral axon guidance in vivo in Caenorhabditis elegans" has been formally accepted for publication in PLOS Genetics! Your manuscript is now with our production department and you will be notified of the publication date in due course.

With kind regards,

Anita Estes

PLOS Genetics

On behalf of:
